# O-GlcNAc on NOTCH1 EGF repeats regulates ligand-induced Notch signaling and vascular development in mammals

**Shogo Sawaguchi[1†], Shweta Varshney[2†], Mitsutaka Ogawa[1†], Yuta Sakaidani[1], Hirokazu Yagi[3], Kyosuke Takeshita[4], Toyoaki Murohara[4], Koichi Kato[3,5], Subha Sundaram[2], Pamela Stanley[2*], Tetsuya Okajima[1*]**

[1]Department of Molecular Biochemistry, Nagoya University Graduate School of Medicine, Nagoya, Japan; [2]Department of Cell Biology, Albert Einstein College of Medicine, New York, United States; [3]Graduate School of Pharmaceutical Sciences, Nagoya City University, Nagoya, Japan; [4]Department of Cardiology, Nagoya University Graduate School of Medicine, Nagoya, Japan; [5]Institute for Molecular Science and Okazaki Institute for Integrative Bioscience, National Institutes of Natural Sciences, Okazaki, Japan

**Abstract** The glycosyltransferase EOGT transfers O-GlcNAc to a consensus site in epidermal growth factor-like (EGF) repeats of a limited number of secreted and membrane proteins, including Notch receptors. In EOGT-deficient cells, the binding of DLL1 and DLL4, but not JAG1, canonical Notch ligands was reduced, and ligand-induced Notch signaling was impaired. Mutagenesis of O-GlcNAc sites on NOTCH1 also resulted in decreased binding of DLL4. EOGT functions were investigated in retinal angiogenesis that depends on Notch signaling. Global or endothelial cell-specific deletion of *Eogt* resulted in defective retinal angiogenesis, with a mild phenotype similar to that caused by reduced Notch signaling in retina. Combined deficiency of different *Notch1* mutant alleles exacerbated the abnormalities in *Eogt*$^{−/−}$ retina, and Notch target gene expression was decreased in *Eogt*$^{−/−}$ endothelial cells. Thus, O-GlcNAc on EGF repeats of Notch receptors mediates ligand-induced Notch signaling required in endothelial cells for optimal vascular development.

**\*For correspondence:** pamela. stanley@einstein.yu.edu (PS); tokajima@med.nagoya-u.ac.jp (TO)

[†]These authors contributed equally to this work

**Competing interests:** The authors declare that no competing interests exist.

## Introduction

N-acetylglucosamine linked to Ser or Thr (O-GlcNAc) is a rare form of post-translational modification specifically modifying epidermal growth factor-like (EGF) domains in secreted or membrane proteins (*Alfaro et al., 2012*; *Matsuura et al., 2008*; *Stanley and Okajima, 2010*). In contrast to O-GlcNAc modification of nuclear and cytoplasmic proteins (*Ma and Hart, 2014*), EGF-specific O-GlcNAcylation occurs in the endoplasmic reticulum (ER) by the action of the EGF-domain-specific O-GlcNAc transferase, EOGT (*Sakaidani et al., 2011*). To date, only a small number of secreted or membrane proteins have been shown to be O-GlcNAcylated, including Notch receptors and their ligands (*Alfaro et al., 2012*; *Müller et al., 2013*; *Tashima and Stanley, 2014*). Notch receptors are also modified by O-fucose and O-glucose glycans at independent consensus sites on EGF repeats (*Figure 1A*), and both types of O-glycan regulate the strength of Notch signaling (*Haltom and Jafar-Nejad, 2015*). Loss of O-fucose, but not O-glucose glycans reduces binding of canonical Notch ligands to Notch receptors, and consequently Notch signaling. Removing one class of O-glycan will affect many EGF repeats but is not expected to affect modification of EGF repeats by unrelated

**eLife digest** As an embryo develops, its cells undergo a series of divisions and transformations until they specialize to form distinct cell types. These transformations are regulated by many molecules and signaling pathways. Often one cell will release signaling molecules that are then recognized by specialized receptor proteins on other cells that then trigger a reaction in the receiving cell. A signaling molecule that binds to a receptor is often referred to as a ligand.

Specific chains of sugar molecules known as glycans are attached to receptor proteins and help to regulate the signaling pathways that control how cells develop in an embryo. One of these receptors is called the Notch receptor, which carries several types of glycans. Mutations that prevent the glycans from being attached to this receptor can lead to congenital diseases. For example, EOGT is an enzyme that attaches a sugar molecule onto the Notch receptor, and mutations in the gene that encodes EOGT are found in people with Adams-Oliver syndrome – a rare condition that causes their skin and limbs to fail to develop properly. However, it was not clear what exact role the attachment of the sugar molecule (referred to as O-GlcNAc) plays and if it regulates Notch signaling.

Sawaguchi et al. studied animal cells grown in the laboratory to investigate ligand binding to Notch receptors and Notch signaling. The experiments showed that if the gene for the EOGT enzyme was deleted, some ligand molecules could not bind well to the Notch signaling receptor. This affected how the receptor became activated and how it triggered a signaling reaction in the receiving cell. In mutant mice that lacked EOGT, the blood vessels of the developing retina had reduced Notch signaling. Blood vessels are important for the normal development of the retina. Together, the results suggest that O-GlcNAc on Notch receptors are important for specific aspects of Notch signaling, including the signals that lead to the normal development of retinal blood vessels.

A next step will be to identify other roles of the O-GlcNAc glycans on Notch receptors and to explore how they affect Notch signaling during development. A better understanding of these mechanisms will extend our knowledge of Adams-Oliver syndrome, which at the moment remains a poorly understood condition.

O-glycans. In this paper, we report that Notch receptors lacking only O-GlcNAc glycans have reduced ligand-induced Notch signaling.

The physiological importance of EOGT was revealed in patients with Adams-Oliver syndrome (AOS), a rare congenital disorder characterized by aplasia cutis congenita and terminal transverse limb defects, often accompanied by cardiovascular malformations and brain anomalies (*Algaze et al., 2013*; *Piazza et al., 2004*). Although the pathogenesis of AOS is broad, it could arise from small-vessel vasculopathy (*Piazza et al., 2004*). AOS is a heterogeneous disorder caused by mutation in one of at least six different genes. Among these, loss-of-function mutations of *EOGT* and *DOCK6* were identified as the basis of an autosomal-recessive form of AOS (*Shaheen et al., 2013*, *2011*). In addition, autosomal dominant mutations of *NOTCH1*, *RBPJ*, *DLL4* and *ARHGAP31* give rise to AOS (*Aminkeng, 2015*; *Hassed et al., 2012*; *Meester et al., 2015*; *Southgate et al., 2015*; *Stittrich et al., 2014*). Gain-of-function mutation of *ARHGAP31* and loss-of-function mutation of *DOCK6* suggested that inactivation of Cdc42/Rac1 functions underlies the molecular basis for AOS. In contrast, loss-of-function mutations of *DLL4*, *RBPJ* and *NOTCH1* in AOS patients suggest that impaired Notch signaling is an alternative basis of the pathogenesis of AOS. Here, we investigate the hypothesis that loss of EOGT affects Notch signaling using cell-based Notch ligand binding and signaling assays and *Eogt* mutant mice. We show that EOGT-catalyzed NOTCH1 O-GlcNAcylation potentiates DLL1- and DLL4-NOTCH1 binding and Notch signaling, whereas JAG1-NOTCH1 binding remains unaffected. Using retinal angiogenesis as a sensitive assay of Notch signaling in vivo (*Roca and Adams, 2007*), we show that mice lacking EOGT have impaired retinal vascular development, with a phenotype characteristic of Notch pathway deficiencies in retina (*Benedito et al., 2009*). Moreover, we show that endothelial functions of EOGT are responsible for the retinal

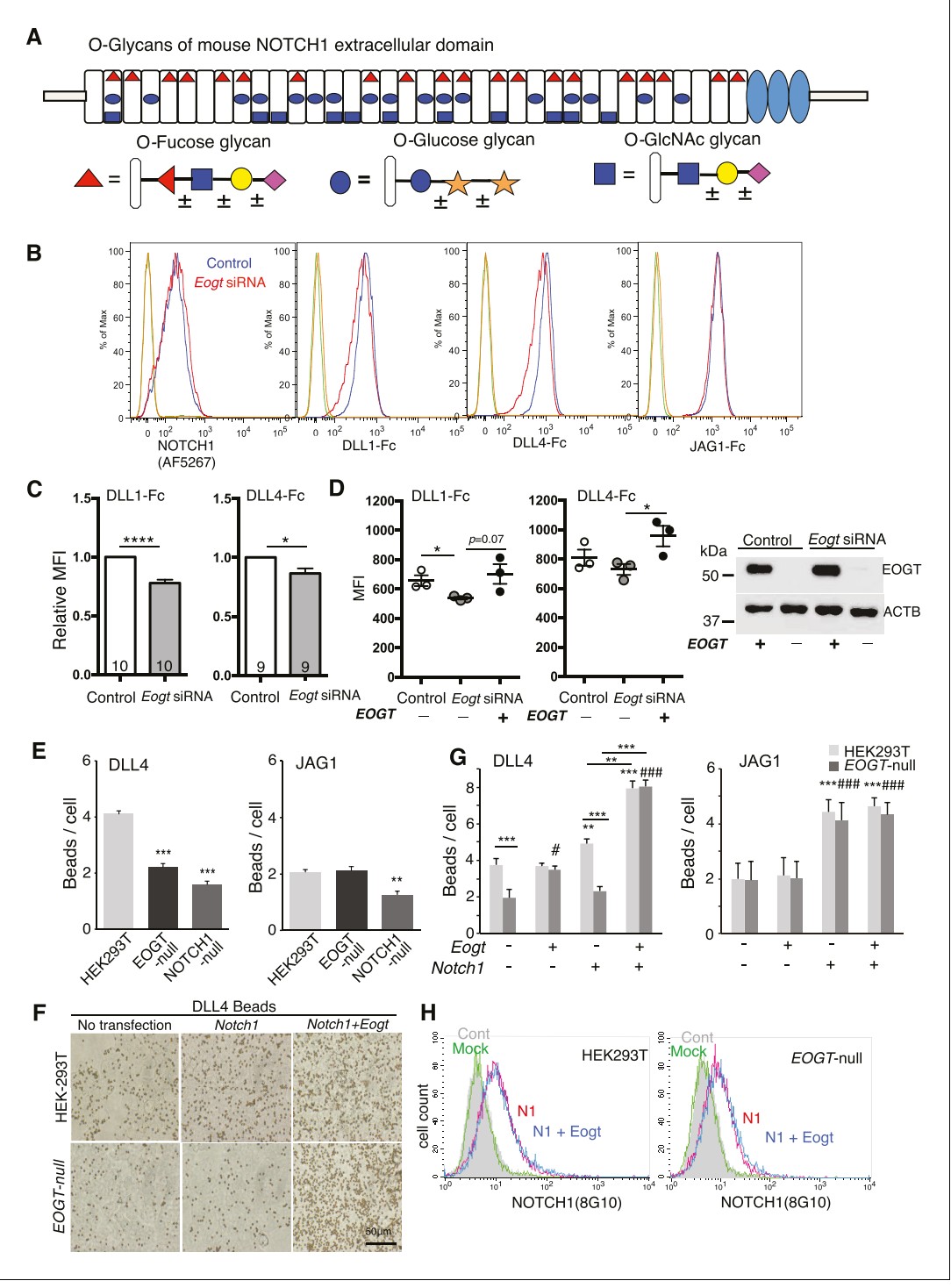

**Figure 1.** EOGT promotes NOTCH1 binding to Delta ligands. (**A**) Diagram of predicted O-glycans on mouse NOTCH1: red triangle, O-fucose glycans; blue circle, O-glucose glycans; blue square, O-GlcNAc glycans. Individual sugar residues that may extend O-fucose, O-glucose or O-GlcNAc to varying degrees are: yellow circle, galactose; pink diamond, sialic acid; orange star, xylose. (**B**) Flow cytometry of Lec1 CHO cells expressing vector control or *Eogt* siRNA with NOTCH1 mAb, DLL1-Fc, DLL4-Fc or JAG1-Fc. (**C**) Relative mean fluorescence index (MFI) for binding of DLL1-Fc and DLL4-Fc to control and *Eogt*-siRNA Lec1 cells. Concentrations of ligand varied from 100 to 750 ng/ml. MFI values for binding to control cells taken as 1.0 were 455 ± 55 (DLL1-Fc) and 1215 ± 49 (DLL4-Fc). Data from 9 to 10 independent experiments are average, normalized MFI ± SEM; significance determined by paired, two-tailed Student's t-test, *p<0.05, ****p<0.0001. (**D**) MFI values obtained for binding of

*Figure 1 continued*

DLL1-Fc or DLL4-Fc (750 ng/ml) to control and *Eogt* knockdown Lec1 CHO cells, before and after transfection of a human *EOGT* cDNA. Data are mean ± SEM from three independent experiments. Significance determined by unpaired, two-tailed Student's t-test, *p<0.05. Western blot analysis of transfectants. (E) DLL4 and JAG1 beads bound to wild-type, *EOGT*-null, or *NOTCH1*-null HEK293T cells were observed by microscopy and counted (n = 50). Data are mean ± S.D. from three independent experiments. Statistical analysis was by Welch's t-test. **p<0.01; ***p<0.001. (F) Wild-type or *EOGT*-null HEK293T cells were transfected with *Notch1* alone or together with *Eogt* followed by incubation with DLL4 beads. The number of DLL4 beads bound to cells was markedly increased by co-transfection of *Eogt* and *Notch1*. (G) Wild-type or *EOGT*-null HEK293T cells or cells transiently transfected with *Notch1* with or without *Eogt*, were incubated with DLL4 or JAG1 beads. The number of Dynabeads bound per transfected cell marked by GFP expression was determined (n = 50). Data are mean ± S.D. from three independent experiments. *p<0.05; **p<0.01; ***p<0.001; #p<0.05; ###p<0.001 compared with the left-most wild type (*) or *EOGT*$^{null}$ (#) bar by Welch's t test. (H) Wild-type or *EOGT*-null HEK293T cells were transfected with *Notch1* with or without *Eogt* and subjected to flow cytometry using 8G10 NOTCH1 Ab. Mock transfectants were analyzed with (*Mock*) or without primary antibody (*Cont*).

The following source data and figure supplement are available for figure 1:

**Source data 1.** Raw data for *Figure 1C,D,E,G*.

**Figure supplement 1.** Generation and characterization of *EOGT*- and *NOTCH1*-null HEK293T cells.

vascular phenotype. Thus, O-GlcNAc on the EGF repeats of Notch receptors is required for optimal Notch signaling in developing retina, and likely in other Notch-dependent processes in mammals.

# Results

## EOGT regulates DLL1 and DLL4 binding to NOTCH1

To address whether EOGT regulates physical interactions between Notch receptors and ligands, Notch ligand binding assays were performed on control and *Eogt*–siRNA Lec1 Chinese hamster ovary (CHO) cells. *Eogt* transcripts determined by quantitative RT-PCR were reduced by ~60%. *Eogt*-siRNA Lec1 cells exhibited reduced binding of soluble DLL1-Fc and DLL4-Fc (*Figure 1B* and *Figure 1C*). However, binding of soluble JAG1-Fc was not altered by knockdown of *Eogt* (*Figure 1B*). Overexpression of an *EOGT* cDNA rescued DLL1 and DLL4 binding (*Figure 1D*). More-over, cell surface expression of NOTCH1 was not reduced in Lec1 cells with reduced *Eogt* (*Figure 1B*).

A second ligand binding assay used soluble Notch ligands attached to Protein A Dynabeads via their Fc domain, and *EOGT*- and *NOTCH1*-null HEK293T cells generated by CRISPR/Cas9 gene editing (*Figure 1—figure supplement 1*). Deletion of *EOGT* was verified using anti-EOGT antibody and by the lack of O-GlcNAc on a NOTCH1 extracellular domain fragment (*Figure 1—figure supplement 1*). Both DLL4 and JAG1 beads/cell were decreased in *NOTCH1*-null cells (*Figure 1E*), confirming that NOTCH1 mediates ligand biding in HEK293T cells. The residual binding capacity of *NOTCH1*-null cells implicated the contribution of other Notch receptors. Similar to *Eogt*-siRNA CHO Lec1 cells, *EOGT*-null HEK293T cells exhibited decreased binding to DLL4 beads, but not to JAG1 beads (*Figure 1E*).

To determine whether overexpression of EOGT potentiates DLL4-NOTCH1 binding, HEK293T cells or *EOGT*-null cells were transfected with *Notch1* and *Eogt* cDNA individually or together, and the ligand binding assay was performed. *Notch1* overexpression led to increased binding of both DLL4 and JAG1 beads to HEK293T cells (*Figure 1F* and *G*). In addition, the effect of *Notch1* overexpression on DLL4 bead binding was selectively impaired in *EOGT*-null cells (*Figure 1F* and *G*). Furthermore, simultaneous expression of *Notch1* and *Eogt* enhanced DLL4 but not JAG1 bead binding, in both HEK293T and *EOGT*-null cells (*Figure 1F* and *G*). The synergistic effect of *Notch1* and *Eogt* on DLL4 bead binding provide strong evidence that EOGT potentiates DLL4-NOTCH1 physical interactions.

As observed in Lec1 CHO cells (*Figure 1B*), neither *Eogt* overexpression nor EOGT loss affected cell surface NOTCH1 expression (*Figure 1H*). Thus, EOGT is not required for NOTCH1 trafficking to the plasma membrane.

## O-GlcNAc on NOTCH1 promotes DLL4-NOTCH1 interactions

To determine whether it is the O-GlcNAc transferred by EOGT to NOTCH1 that directly affects the binding of DLL4, we generated NOTCH1 site-specific mutants by Ala substitution of Ser/Thr in predicted O-GlcNAcylation sites. Alignment of previously reported O-GlcNAc-modified proteins including Notch and Dumpy in *Drosophila*, and Hspg2, Nell1, Lama5, and Pamr1 in mouse brain suggested that the potential consensus sequence is $C^5XXG(Y/F/L)(T/S)GX_{2-3}C^6$ (*Alfaro et al., 2012*; *Sakaidani et al., 2011*). A relatively small number of the EGF domains in mouse NOTCH1 contain the consensus sequence $C^5XXG(Y/F)(T/S)GXXC^6$, and these are conserved among mouse, human, rat, Chinese hamster and zebrafish in EGF10, 14, 15, 17, 20, 23, 26, 27 and 29 (*Figure 2—figure supplement 1*). Mutation of two or four Ser/Thr to Ala in EGF2, 10, 17 and 20 ($Notch1^{\Delta2xO-GlcNAc}$; $Notch1^{\Delta4xO-GlcNAc}$; *Figure 2A*), outside the canonical ligand-binding region (EGF11 and 12), was performed. Cell surface expression of the NOTCH1 mutants was indistinguishable from wild-type NOTCH1 (*Figure 2B*), consistent with unchanged NOTCH1 cell surface expression in *EOGT*-null HEK293T cells (*Figure 1H*). Decreased O-GlcNAcylation of $Notch1^{\Delta2xO-GlcNAc}$ and $Notch1^{\Delta4xO-GlcNAc}$ mutants was confirmed by immunoblotting with CTD110.6 O-GlcNAc antibody (*Figure 2C*). Expression of $Notch1^{\Delta4xO-GlcNAc}$ in *Eogt*-transfected cells resulted in an ~80% decrease in O-GlcNAc immunostain signal (*Figure 2D*). By contrast, Ala substitution in only EGF2 and 10 ($Notch1^{\Delta2xO-GlcNAc}$) removed ~60% of the signal in *Eogt* transfectants (*Figure 2D*). Moreover, the number of DLL4 beads bound to NOTCH1$^{\Delta4xO-GlcNAc}$ transfectants was significantly decreased relative to wild-type NOTCH1 (*Figure 2E*), similar to the decrease observed in *EOGT*-null cells (*Figure 1E*). This suggests that O-GlcNAc in EGF2, 10, 17 and/or 20 are important contributors to DLL4/NOTCH1 interactions. When NOTCH1$^{\Delta4xO-GlcNAc}$ was simultaneously expressed with *Eogt*, $Notch1^{\Delta4xO-GlcNAc}$/*Eogt* cotransfectants also exhibited impaired binding to DLL4 beads. These results demonstrate that DLL4/NOTCH1 interactions mediated by EOGT require O-GlcNAc on sites that are located outside the canonical ligand-binding region. In contrast, the number of JAG1 beads bound to *Notch1* transfectants was not reduced in $Notch1^{\Delta4xO-GlcNAc}$ transfectants, irrespective of *Eogt* overexpression (*Figure 2E*). These results provide strong evidence that O-GlcNAc on NOTCH1 EGF2, 10, 17 and/or 20 selectively affect DLL4/NOTCH1 but not JAG1/NOTCH1 physical interactions. The substantial decrease in O-GlcNAc signal following removal of only four of the nine potential O-GlcNAcylation sites, suggests that a limited number of sites are O-GlcNAcylated in NOTCH1.

## Knockdown of EOGT reduces Notch signaling

To investigate effects of EOGT on Notch ligand-induced signaling, we analyzed NOTCH1 activation and signaling in HeLa and Lec1 CHO cells with reduced *Eogt*. HeLa cells stably expressing four different shRNA constructs targeted to the coding or 3'UTR region of human *EOGT* were co-cultured with L cells, or DLL1-expressing L cells (D1/L), in the presence and absence of a gamma-secretase inhibitor (GSI). Ligand-induced activation of NOTCH1 generates the release of NOTCH1 intracellular domain (ICD), identified by Western analysis using mAb Val1744 specific for the cleaved N-terminus of NOTCH1 ICD (*Huppert et al., 2000*). NOTCH1 cleavage was stimulated poorly, or not at all, by L cells (*Figure 3A*). However, co-culture with D1/L cells caused robust NOTCH1 activation in cells expressing a *GAPDH* control shRNA (*Figure 3A*). Importantly, NOTCH1 cleavage was inhibited in D1/L co-cultures containing the GSI (*Figure 3A*). All four *EOGT*-targeted shRNA constructs specifically reduced NOTCH1 activation. These findings were replicated and also reproduced in an independent set of HeLa cell transductants. Because endogenous levels of EOGT were not readily detectable in HeLa cells by Western analysis, and reduced transcript levels do not reflect enzyme activity, knockdown efficiency was determined by Western blot detection of the O-GlcNAc product of EOGT. The TA197 and TA198 shRNAs that were most effective at reducing NOTCH1 activation (*Figure 3A*), exhibited marked loss of O-GlcNAc on species comigrating with NOTCH1 (*Figure 3B*; the lower band of the O-GlcNAc-positive doublet). HeLa cells express NOTCH2 and NOTCH3, which may also be modified by O-GlcNAc. Alternatively, the O-GlcNAc mAb may detect a glycoform of NOTCH1 that is not detected by the NOTCH1 extracellular domain (ECD) mAb. Image J analysis of

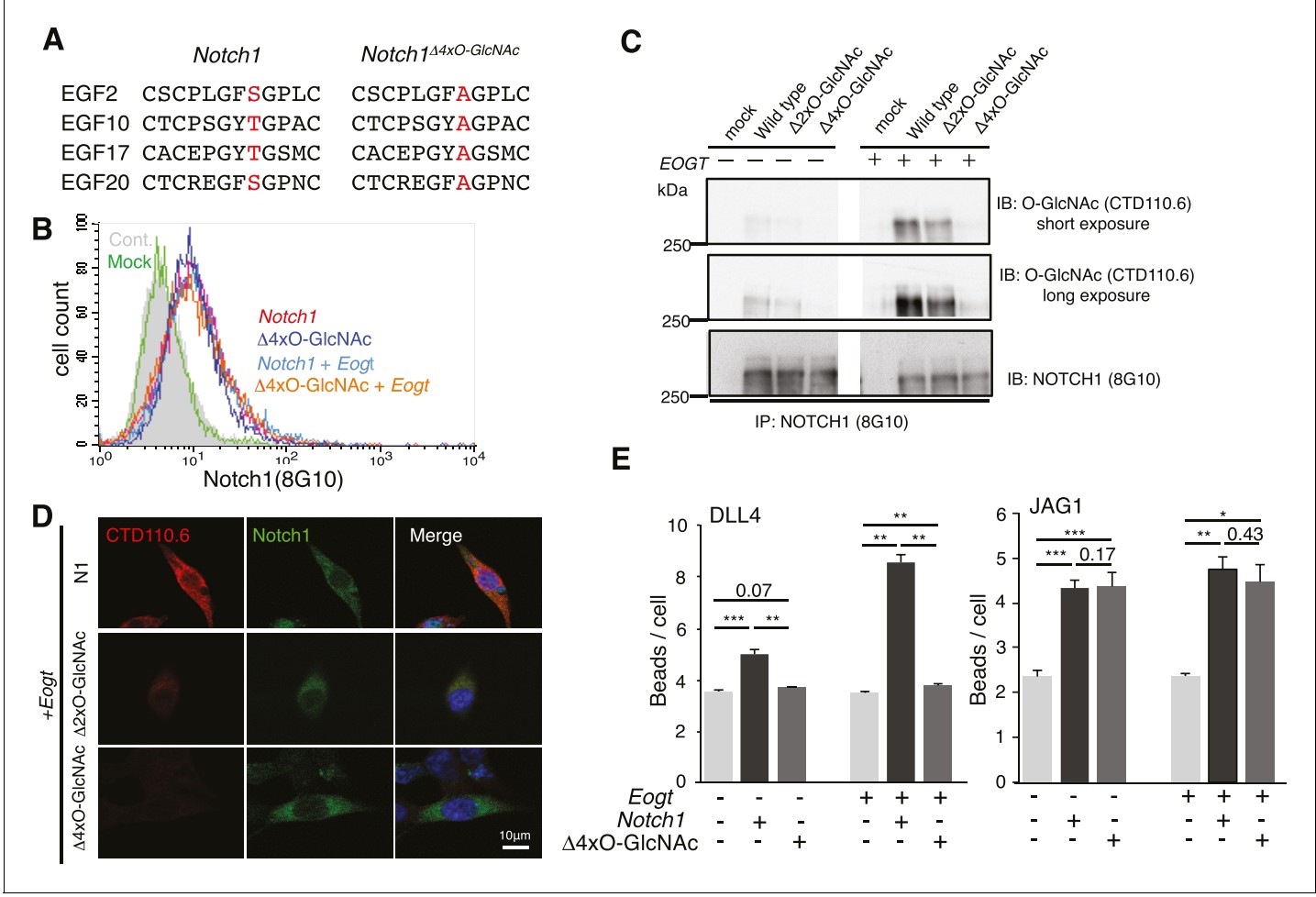

**Figure 2.** O-GlcNAc on NOTCH1 EGF repeats promotes DLL4-NOTCH1 interactions. (**A**) Ala substitution of Thr/Ser in the O-GlcNAc consensus site $C^5XXG(Y/F)(T/S)GXXC^6$ in EGF2, 10, 17, and 20 in the NOTCH1$^{\Delta4xO-GlcNAc}$ mutant. (**B**) Cell surface expression of NOTCH1$^{\Delta4xO-GlcNAc}$ is comparable to that of wild-type NOTCH1. *EOGT*-null transfectants expressing *Notch1* or *Notch1*$^{\Delta4xO-GlcNAc}$ with or without *Eogt* were subjected to flow cytometry using 8G10 NOTCH1 antibody. Control transfectants were with (*Mock*) or without (*Cont*) primary antibody. Similar results were obtained for wild-type HEK293T cells (not shown). (**C**) Expression and O-GlcNAcylation of *Notch1*, *Notch1*$^{\Delta2xO-GlcNAc}$ and *Notch1*$^{\Delta4xO-GlcNAc}$ were analyzed by immunoprecipitation (IP) using NOTCH1 (8G10) antibody, followed by immunoblotting with CTD110.6 O-GlcNAc or NOTCH1 antibodies. (**D**) HEK293T transfectants expressing NOTCH1, NOTCH1$^{\Delta2xO-GlcNAc}$ or NOTCH1$^{\Delta4xO-GlcNAc}$ were immunostained for O-GlcNAc (CTD110.6 mAb; red) or NOTCH1 (8G10 mAb; green). Merged images include DAPI (blue) staining. Note that CTD110.6 mAb binding is markedly decreased in the NOTCH1$^{\Delta4xO-GlcNAc}$ mutant. (**E**) Quantification of the number of DLL4- or JAG1-coated Dynabeads bound to *Notch1* versus *Notch1*$^{\Delta4xO-GlcNAc}$ transfected cells (marked with GFP), in the presence and absence of *Eogt*. Data are mean ± S.D from three independent experiments. Each experiment analyzed 50 cells. *p<0.05; **p<0.01; ***p<0.001 (Welch's t test).

The following source data and figure supplement are available for figure 2:

**Source data 1.** Raw data for *Figure 2E*.

**Figure supplement 1.** NOTCH1 O-GlcNAc site mutants.

immunoprecipitated *Notch1*-MycHis using CTD110.6, NOTCH1 and Myc mAbs confirmed the reduction in O-GlcNAc on NOTCH1 in *EOGT*-knockdown HeLa cells. To investigate specificity, HeLa cells expressing *GAPDH* or TA197 shRNA against the 3'UTR of *EOGT*, were transfected with empty vector or a human *EOGT* cDNA, and DLL1-induced activation of NOTCH1 was determined. Overexpression of *EOGT* increased the amount of activated NOTCH1 induced by D1/L cells in control HeLa cells, including in the presence of the GSI (*Figure 3C*, *D*). Partial rescue of *EOGT* knockdown by

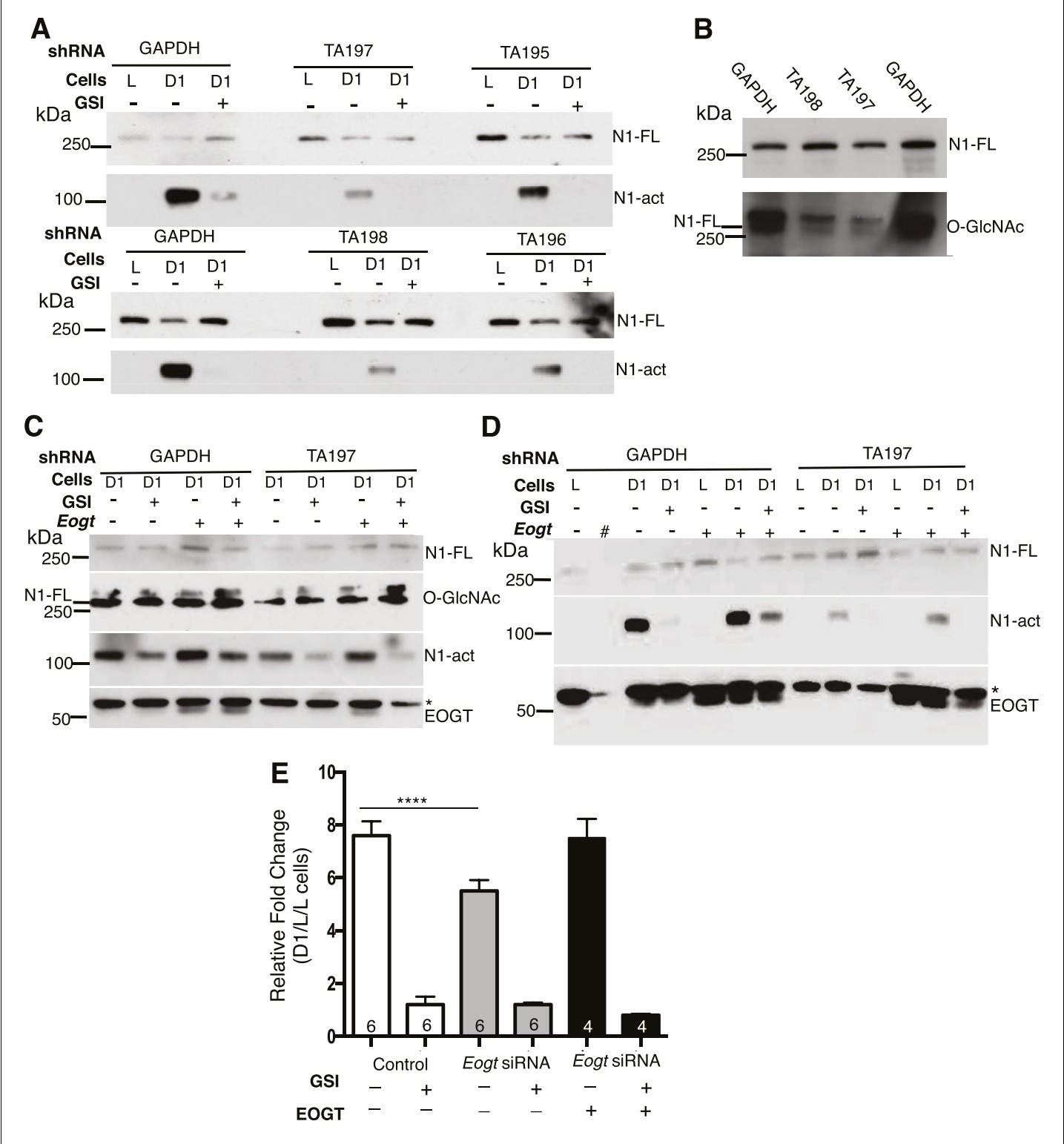

**Figure 3.** Notch signaling is reduced in EOGT-deficient cells. (**A**) Knockdown of *EOGT* inhibits NOTCH1 activation cleavage. HeLa cells stably expressing shRNAs targeting *GAPDH* or *EOGT* (TA195, TA196, TA197, TA198) were co-cultured with L cells or D1/L (D1) cells in the presence and absence of 1 µM DAPT (GSI). After 6 hr, lysates were subjected to Western blot analysis using Abs to detect activated NOTCH1 (N1-act) and NOTCH1 full length (N1–FL) on the relevant section of the PVDF membrane. (**B**) Western blot analysis of samples from (**A**) using Ab to detect O-GlcNAc, followed after stripping by Ab to detect N1-FL. (**C**) HeLa cells stably expressing shRNA against *GAPDH* or *EOGT* (TA197) were transfected with vector control or a human *EOGT* cDNA. After 4 days, co-culture was performed with D1/L (D1) cells in the presence and absence of the DAPT. After ~7 hr,

*Figure 3 continued on next page*

Figure 3 continued

lysates were subjected to Western analysis to detect N1-FL, N1-act and EOGT on the relevant section of the PVDF membrane. O-GlcNAc was detected after stripping the N1-FL membrane section. * non-specific band. (D) The same as (C) except co-culture was with L cells or D1/L cells (D1). The second lane (#) was left empty. * non-specific band. (E) Knockdown of *Eogt* reduces ligand-induced Notch signaling. Lec1 CHO cells stably expressing siRNAs targeted against *Eogt* were transfected with TP1-luciferase and TK-renilla luciferase, co-cultured for 30 hr with L cells or D1/L cells, with and without GSI IX (12.5 μM) or a human *EOGT* cDNA, and dual firefly and renilla luciferase assays were performed. Normalized firefly luciferase activity in L versus D1/L cell co-cultures was plotted as fold-change. The number of independent experiments, each performed in duplicate, are shown in the histogram. Error bars are mean ± standard error and p values were determined by two-tailed paired Student's t-test (****p<0.0001).

The following source data is available for figure 3:

**Source data 1.** Raw data for *Figure 3E*.

TA197 (and TA198, not shown) was achieved by a human *EOGT* cDNA (*Figure 3C*, *D*; representative of three independent experiments).

Ligand-induced Notch signaling was also investigated in CHO Lec1 *Eogt*-siRNA cells (*Tashima and Stanley, 2014*) using the NOTCH1 ICD-responsive TP1-luciferase reporter. Reduction of *Eogt* correlated with reduced Notch signaling (*Figure 3E*) and transfection of a human *EOGT* cDNA largely rescued signaling in the *Eogt* knockdown cells. Taken together, the combined data provide strong evidence that EOGT regulates DLL1-induced Notch signaling.

## Generation of *Eogt*-targeted mice

To investigate biological functions of O-GlcNAc on EGF repeats in vivo, *Eogt* mutant mice were generated. A floxed *Neo* allele (*Eogt*$^{flNeo}$) with *lox*P sites flanking exon 10 and a Neo cassette flanked by *lox*P/FRT sites was constructed (*Figure 4A*). Southern blotting showed successful homologous recombination in an *Eogt*$^{flNeo}$ mouse (*Figure 4B*). *Eogt*$^{flNeo}$ mice were crossed to Flp-deleter mice to obtain an *Eogt*$^F$ mouse (*Figure 4A*). Alternatively, *Eogt*$^{flNeo}$ mice were crossed to global Cre-deleter mice to delete exon 10 (*Figure 4C*). The amino acid sequence encoded by exon 10 constitutes part of the putative catalytic domain, and Cre recombinase-mediated excision of the sequence between the *lox*P sites would cause a frame-shift mutation, generating a catalytically inactive protein (*Figure 4—figure supplement 1*). RT-PCR detected low levels of aberrant transcripts lacking exon 10 (*Figure 4D*) and quantitative RT (qRT)-PCR confirmed that *Eogt* transcripts were low, possibly due to nonsense-mediated mRNA decay (*Figure 4E*). Therefore, the *Eogt* gene lacking exon 10 is effectively a null allele. Indeed, immunoblotting confirmed the absence of EOGT protein in the lungs of *Eogt*$^{-/-}$ mice (*Figure 4F*).

Previous studies revealed that *Eogt* is widely expressed in various tissues in the mouse, although specific cell types expressing *Eogt* were not identified (*Sakaidani et al., 2012*). We now show by in situ hybridization that *Eogt* is expressed in endothelial cells (EC) of retinal arteries, capillaries and veins at P5, and mainly in arteries and veins at P15 (*Figure 4G*). At the P5 vascular front, *Eogt* signal was apparent in tip cells (*Figure 4G*). These signals were absent in *Eogt*$^{-/-}$ retinas and control retinas stained with sense probes (*Figure 4—figure supplement 2A*). To prove EC expression of *Eogt*, in situ hybridization was performed in retinas from conditional mutant mice in which *Eogt* expression was specifically eliminated in ECs. *Tek-Cre;Eogt*$^{F/F}$ retinas exhibited a marked decrease in vascular staining with the *Eogt* probe (*Figure 4H* and *Figure 4—figure supplement 2B*). Consistently, double staining showed that *Eogt* signal partially overlaps with the signal from isolectin-B4 (IB4) that stains ECs (*Figure 4H*), confirming *Eogt* expression in ECs. Residual staining in tissue surrounding the retinal vasculature suggested weaker *Eogt* expression in neuronal tissues.

## *Eogt* is required for optimal retinal angiogenesis

*Eogt*$^{-/-}$ mice were obtained at the expected Mendelian ratio of ~25% (total n = 244), and did not exhibit obvious abnormalities. Since *Eogt* is highly expressed in ECs, and retinal angiogenesis requires Notch signaling (*Roca and Adams, 2007*), we focused our analysis on angiogenesis and vessel formation in the postnatal mouse retina. Retinal angiogenesis is induced just after birth in mice when a single layer of superficial retinal plexus grows from the center toward the periphery until P7. Although a vascular plexus was formed in P5 *Eogt*$^{-/-}$ retina, vascular progression toward

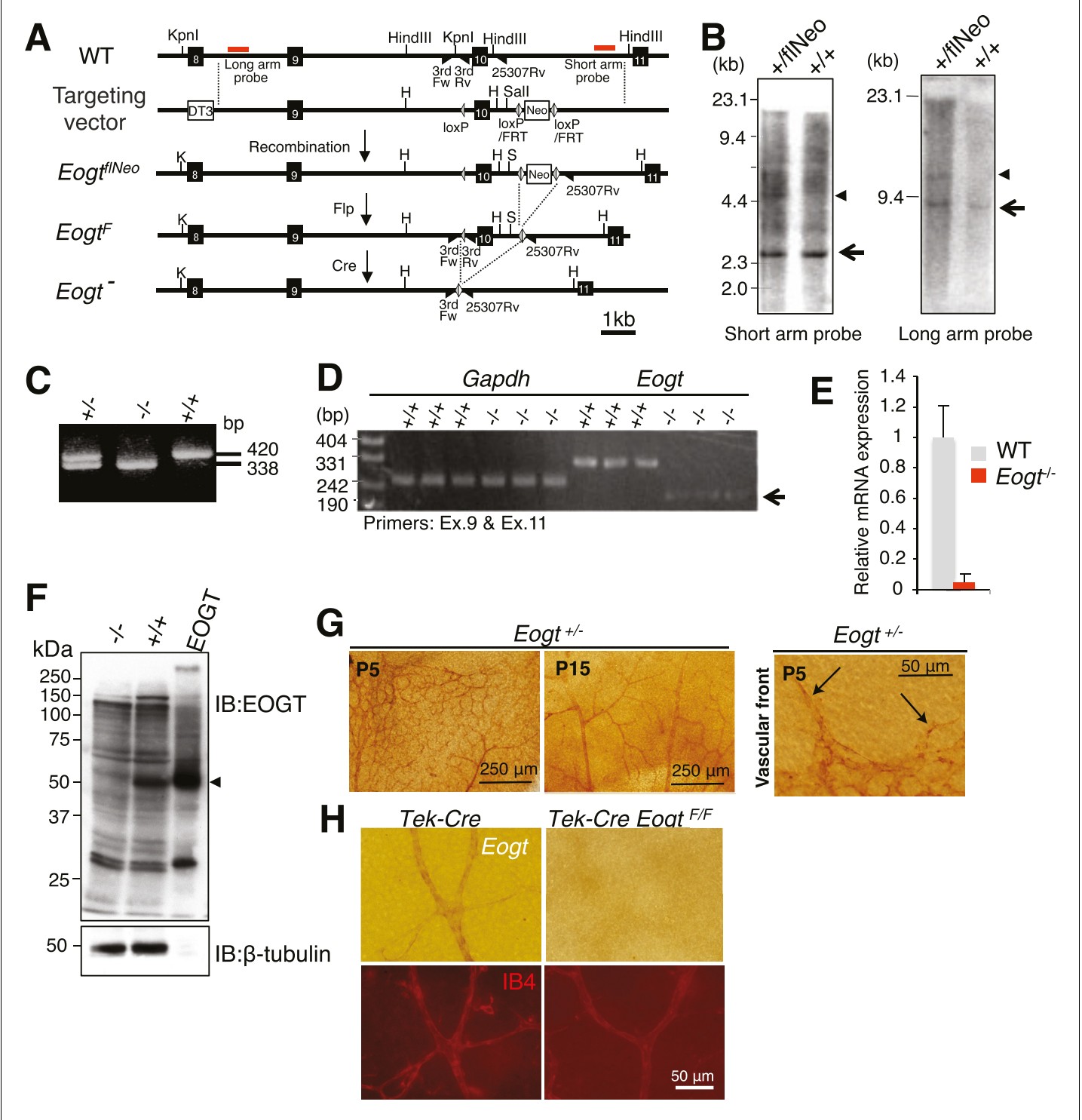

**Figure 4.** Generation of *Eogt*-targeted mice. (**A**) Schematic drawing of the wild-type mouse *Eogt* allele (WT), the targeting vector, the floxed allele with the neomycin (Neo)-resistance gene (*Eogt*<sup>flNeo</sup>), the floxed allele without Neo (*Eogt*<sup>F</sup>), and the deleted allele (*Eogt*<sup>−</sup>). *Eogt* exons (closed boxes); Neo and diphtheria toxin (DT3) genes served as positive and negative selection markers, respectively (open boxes); *lox*P sites (gray triangles); FRT sites (open triangles); *Kpn*I (K), *Hind*III (H), and *Sal*I (S) restriction sites. Homologous recombination between the WT allele and the targeting vector generated the *Eogt*<sup>flNeo</sup> allele. The *Eogt*<sup>F</sup> and *Eogt*<sup>−</sup> alleles were obtained by Flp-mediated and Ayu1-Cre-mediated recombination, respectively. Red lines indicate positions of probes used for Southern blotting. Positions of primers used for genotyping are indicated by closed triangles. (**B**) *Hind*III- or *Kpn*I/*Sal*I-digested genomic DNA isolated from WT (+/+) or heterozygous floxed neo (+/flNeo) ES cells was analyzed by Southern blotting using the short arm or long arm probe, respectively. In addition to signal from the WT allele (arrow), the flNeo mouse shows an additional band corresponding to

*Figure 4 continued on next page*

*Figure 4 continued*

the recombinant allele (arrowhead). (C) Genomic DNA isolated from WT, *Eogt*[+/−], and *Eogt*[−/−] mice was subjected to genotyping using 3rdloxFw, 3rdloxRv, and 25307Rv primers. (D) Semi-quantitative RT-PCR analysis of total RNA from WT or *Eogt*[−/−] brain ECs using primers targeting exons 9 and 11. Minor transcripts lacking exon 10 were detected in *Eogt*[−/−] mice (arrow). *Gapdh* was amplified as an internal control. (E) Quantification of *Eogt* transcripts in *Eogt*[−/−] and WT mouse. qRT-PCR analysis of brain ECs showed a marked decrease in *Eogt* transcripts. Data are mean ± S.D. from three independent experiments performed in triplicate. Each experiment analyzed pooled total RNA obtained from 10 mice. (F) Lack of EOGT protein expression in *Eogt*[−/−] mouse. Lung lysates prepared from adult WT or *Eogt*[−/−] mice were analyzed in parallel with cell lysates from HEK293T cells overexpressing *Eogt*. Immunoblotting was performed using anti-EOGT or β-tubulin antibodies. (G) Whole-mount in situ hybridization for *Eogt* in the P5 or P15 retina. Vascular staining of *Eogt* is evident in *Eogt*[+/−] retina. *Eogt* expression in vascular sprouts is indicated by arrows. (H) P5 control *Tek*-Cre and *Tek*-Cre *Eogt*[F/F] retinas were subjected to in situ hybridization. Counter staining with Dylight 594-conjugated IB4 is shown below.

The following source data and figure supplements are available for figure 4:

**Source data 1.** Raw data for *Figure 4E*.

**Figure supplement 1.** Deletion of exon 10 in the *Eogt* gene causes exon nine to be spliced to exon 11 leading to a frame shift encoding six novel amino acids before a stop codon.

**Figure supplement 2.** Whole-mount in situ hybridization for *Eogt*.

the periphery was delayed (*Figure 5A*). Vessel maturation measured by the association of blood vessels with mural cells was compromised, as evident by the reduced length of αSMA-positive vessels in P5 *Eogt*[−/−] retinas (*Figure 5A*). However, the distribution of anti-NG2-positive pericytes was unaffected at P5 in *Eogt*[−/−] retinas (data not shown). By contrast, the density and number of branch points were increased in P5 *Eogt*[−/−] retinas (*Figure 5B*). At the vascular front, the number of filopodia was also increased in P5 *Eogt*[−/−] retinas (*Figure 5C*). An increase in vascular branching was also observed in P15 *Eogt*[−/−] retinas (*Figure 5D*). While some variation in retinal angiogenesis was observed in heterozygote *Eogt*[+/−] retinas, the phenotype was not autosomal dominant. These data suggest that EOGT is required for optimal retinal vascular development. Moreover, the phenotype exhibited by *Eogt*[−/−] retinas is similar to, although weaker than, that observed in retinas of Notch pathway mutants (*Benedito et al., 2009*; *Kofler et al., 2015*; *Phng and Gerhardt, 2009*).

Loss of Notch signaling specifically in ECs also results in an increase in vascular branching (*Hellström et al., 2007*; *Leslie et al., 2007*; *Lobov et al., 2007*; *Ridgway et al., 2006*; *Siekmann and Lawson, 2007*; *Suchting et al., 2007*). To investigate *Eogt* functions in ECs, retinas with EC-specific deletion of *Eogt* were examined. In both P5 and P15 retinas, vessel branching was increased in *Tek-Cre:Eogt*[F/F] retinas compared with control *Tek-Cre* retinas (*Figure 5—figure supplement 1*). Therefore, EOGT in ECs regulates vascular branching during retinal angiogenesis. Interestingly, however, the length of αSMA-positive vessels and vessel progression in P5 *Tek-Cre:Eogt*[F/F] retinas were not significantly changed compared with *Tek-Cre* control retinas (data not shown). Since Notch signaling in pericytes and macrophages also regulates retinal angiogenesis (*Kofler et al., 2015*; *Outtz et al., 2011*), this result suggests additional *Eogt* functions in other cell types that were not affected in the EC-specific knockout of *Eogt*.

## Loss of *Eogt* phenocopies *Notch1* haploinsufficiency and compound mutants display an enhanced vascular phenotype

To investigate whether EOGT interacts with NOTCH1 to control retinal angiogenesis, we determined the effects of decreasing NOTCH1 or Notch signaling on the retinal vascular phenotype in an *Eogt*-null background. We chose mutant alleles of *Notch1* and *Rbpj* for these analyses because their haploinsufficiency causes AOS, as does mutant *EOGT* homozygosity. As expected from previous reports (*Hellström et al., 2007*; *Kofler et al., 2015*), *Notch1* heterozygous retinas showed increased vessel branching at P5, and a higher number of filopodia (*Figure 6A,B and C*), similar to that observed in retinas of *Eogt*-null mice (*Figure 5*). These phenotypes were exaggerated in an *Eogt*[−/−] background. Enhanced vascular defects in *Eogt*[−/−]*Notch1*[+/−] compound mutant retinas were also observed in P15 retinas (*Figure 6D and E*). Similar to Notch1 heterozygotes, *Rbpj*[+/−] retinas exhibited elevated vessel branching at P5, which was further enhanced in *Eogt*[−/−]*Rbpj*[+/−] compound

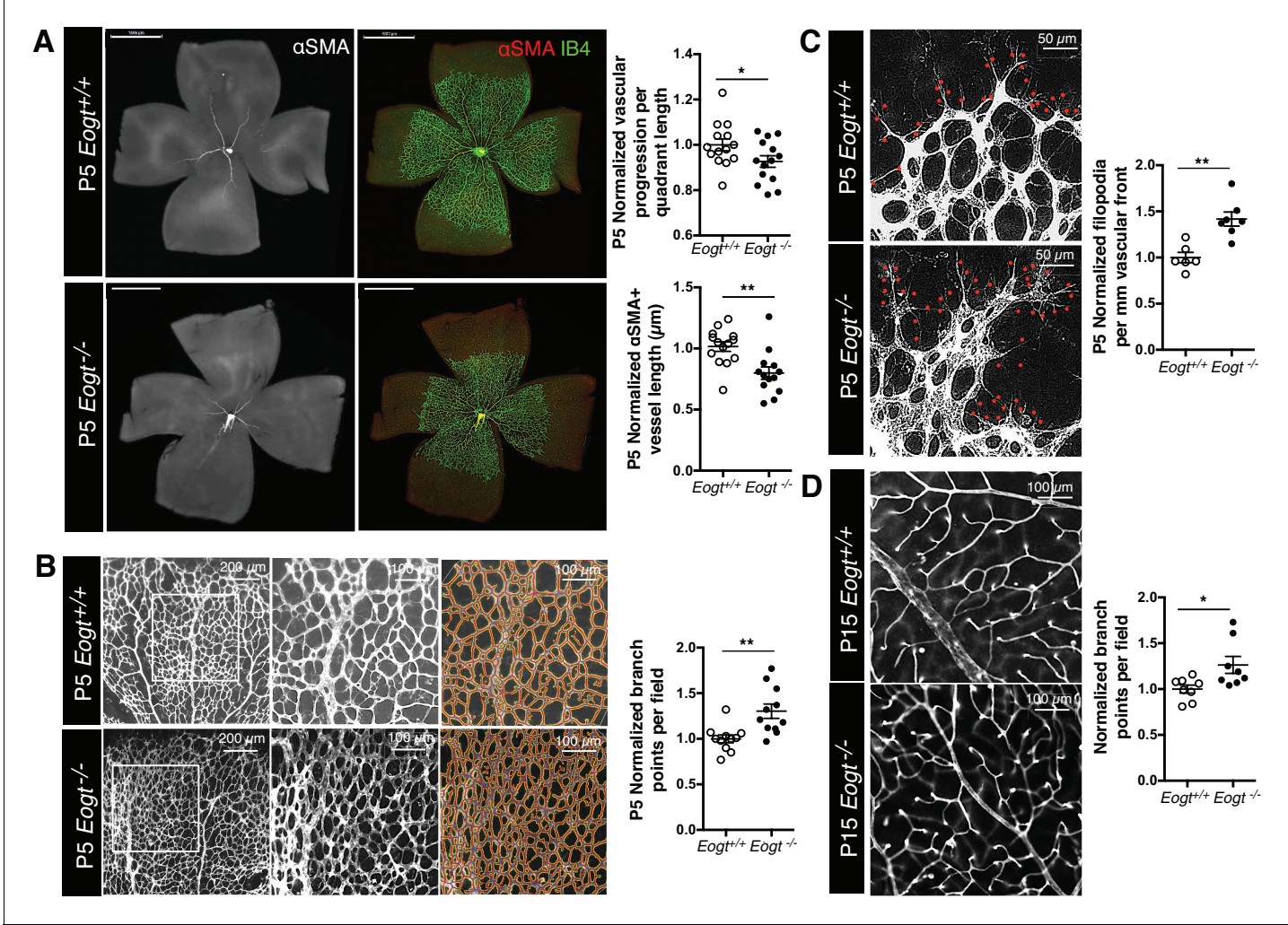

**Figure 5.** EOGT regulates retinal angiogenesis. (A) Whole-mount images of *Eogt*$^{+/+}$ or *Eogt*$^{-/-}$ P5 retinas stained with IB4 and anti-αSMA antibody. Bars represent 1000 μm. Scatter plots at right show vascular progression per quadrant length, per retina, and αSMA$^+$ vessel length from the optic nerve per retina, normalized to *Eogt*$^{+/+}$ retinas (taken as 1.0). Each symbol represents average vessel progression from 3 to 4 quadrants, or the average length from the optic nerve of 3–4 αSMA$^+$ vessels, per retina. For *Eogt*$^{+/+}$, average vascular progression was 0.63 ± 0.03, and the average αSMA$^+$ vessel length was 962 ± 58 μm. (B) Images of P5 vascular front of *Eogt*$^{+/+}$ or *Eogt*$^{-/-}$ retinas stained with IB4. Scatter plot at right shows the normalized number of vascular branch points (500 × 500 μm field (n = 3–8 fields per retina), N = 11 mice). Each symbol represents the average per retina, per mouse. The average *Eogt*$^{+/+}$ branch points were 302 ± 27 per retina, taken as 1.0 for normalization. (C) Images of filopodia emanating from tip cells (red dots) at the vascular front. The scatter plot at right shows the number of filopodia in P5 retinas (250 × 250 μm field (n = 4–12 fields per retina), N = 6 mice). The average filopodia per mm vascular front for *Eogt*$^{+/+}$ was 33 ± 2 per mm, taken as 1.0 for normalization. (D) Images of P15 retinas stained with IB4. The scatter plot at right shows number of branch points in P15 retinas (500 × 500 μm field (n = 3–8 fields per retina), N = 8 mice). The average for *Eogt*$^{+/+}$ was 74 ± 3 per field. Data were normalized from mean ± standard error; p values were calculated by unpaired two-tailed Student's TTEST. *p≤0.05; **p≤0.01; ***p≤0.001.

The following figure supplement is available for figure 5:

**Figure supplement 1.** Images of control *Tek*-Cre and *Tek*-Cre *Eogt*$^{F/F}$ retinas stained with isolectin B4 (IB4) and quantification of branch points in P5 (N = 6 mice) and P15 (N = 3 mice) retinas and numbers of filopodia per mm vascular front in P5 retinas (N = 6 mice).

mutant retinas (*Figure 6A and B*). These results are consistent with EOGT and Notch signaling acting in the same or parallel pathways. However, heterozygotes expressing the hypomorphic alleles *Notch1*$^{12f}$ and *Notch1*$^{lbd}$ (*Ge and Stanley, 2008*, *2010*) did not have a retinal phenotype and yet,

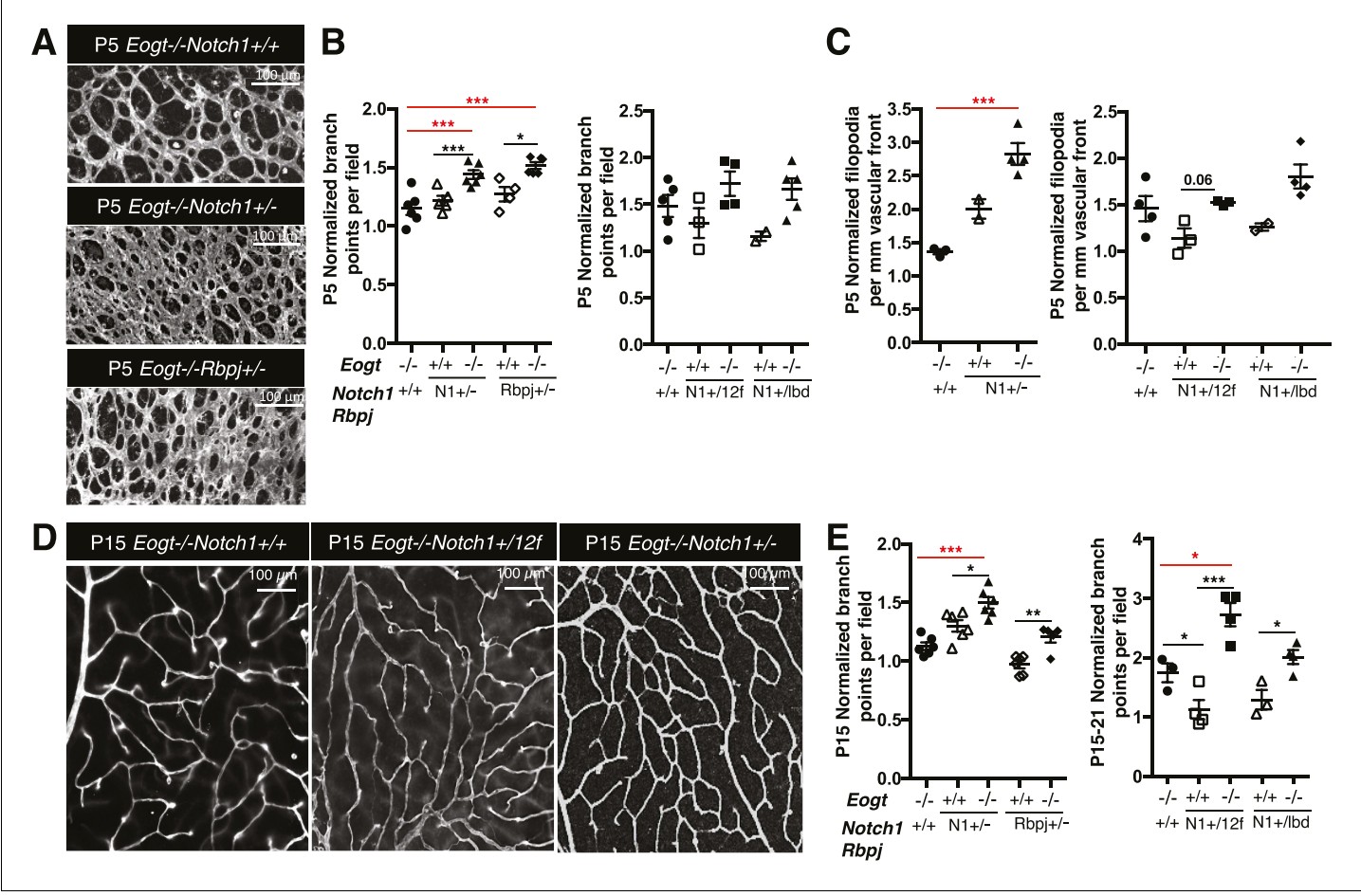

**Figure 6.** Reduced Notch signaling augments the loss of *Eogt* in retinal angiogenesis. (**A**) Images of vessel density in P5 retinas from *Eogt*$^{-/-}$ compared to compound mutant mice. (**B**) Scatter plots represent branch point numbers in P5 retinas from *Eogt*$^{-/-}$ compared with compound mutant mice, normalized to *Eogt*$^{+/+}$ mice. The average number of branch points for the *Eogt*$^{+/+}$ used to compare mice with an N1$^-$ or Rbpj$^-$ allele was 237 ± 5, and for the *Eogt*$^{+/+}$ compared to mice with an N1$^{12f}$ or N1$^{lbd}$ allele was 380 ± 36 (500 X 500 μm field (n = 3–8 fields per retina), N = 2–6 mice). (**C**) Scatter plots showing the number of filopodia in P5 compound mutant mice as compared to *Eogt*$^{-/-}$ mice, normalized to *Eogt*$^{+/+}$ mice. Each symbol represents the average number of filopodia per mouse (250 × 250 μm field (n = 4–12 fields per mouse), N = 2–4 mice. The average for *Eogt*$^{+/+}$ compared to mice with a N1$^-$ or Rbpj$^-$ allele was 30 ± 1 per mm and for mice with a N1$^{12f}$ or N1$^{lbd}$ allele was 36 ± 4 per mm, taken as 1.0 for normalization. (**D**) Images of branch points in P15 retinas comparing *Eogt*$^{-/-}$ to compound mutant mice as indicated. (**E**) Scatter plots shows branch points in compound mutants compared to *Eogt*$^{-/-}$ P15 retinas, normalized to *Eogt*$^{+/+}$ mice. The average of the *Eogt*$^{+/+}$ used to compare mice with a with a N1$^-$ or Rbpj$^-$ allele was 73 ± 3, and for mice with a N1$^{12f}$ or N1$^{lbd}$ allele it was 69 ± 12 (500 x 500 μm field (n = 3–8 fields per retina), N = 3–6 mice. Data represent mean ± standard error except for P5 *Eogt*$^{+/+}$*Notch1*$^{+/-}$ filopodia and P5 *Eogt*$^{+/+}$*Notch1*$^{+/lbd}$ branch points and filopodia that are represented as mean ± range; p values were calculated by unpaired two-tailed Student's t-test. *p≤0.05; **p≤0.01; ***p≤0.001.

they augmented the *Eogt*$^{-/-}$ phenotype at P5 and P15, respectively (*Figures 6B, C, D and E*). The enhancement in the *Eogt*$^{-/-}$ retinal vascular phenotype in two different *Notch1* hypomorphic backgrounds that do not themselves exhibit retinal defects, strongly suggests that the addition of O-GlcNAc by EOGT is important in regulating the Notch signaling pathway.

## EOGT is required for retinal vascular integrity

Notch is an essential contributor to vascular integrity (*Hofmann and Iruela-Arispe, 2007*). Increased vascular permeability is manifested by extravasation of plasma proteins such as plasminogen, fibrinogen and albumin. Immunostaining of P15 wild-type retina with anti-fibrinogen antibody detected sporadic signals in the vascular area, whereas no extravascular signal was observed. In contrast, *Eogt*$^{-/-}$ retina exhibited diffuse fibrinogen staining outside blood vessels, and prominent staining

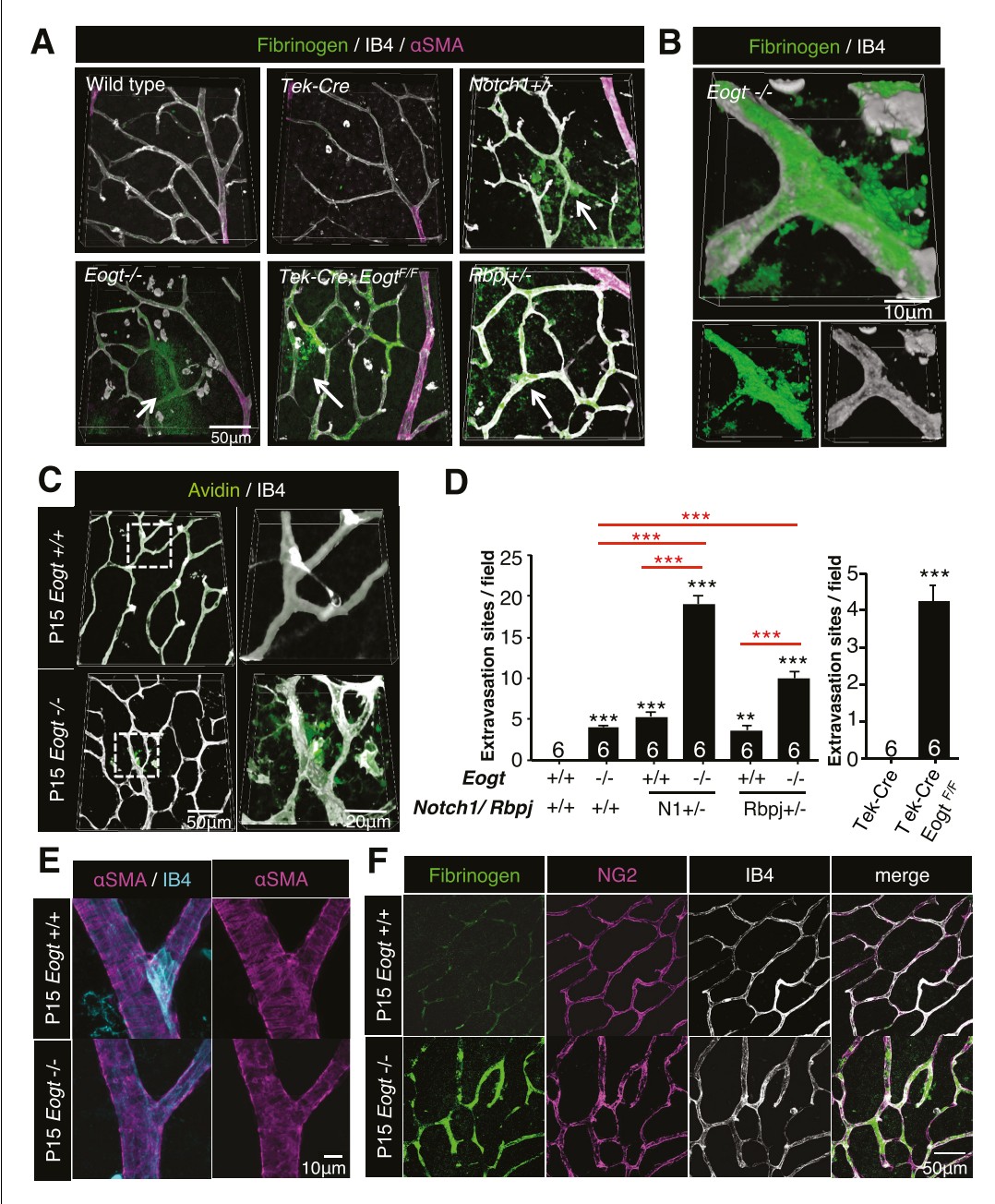

**Figure 7.** Reduced vessel integrity in the *Eogt*$^{-/-}$ retina. (**A**) Immunostaining with fibrinogen (green) and α-SMA (magenta) antibodies in P15 wild-type, *Eogt*$^{-/-}$, *Tek-Cre*, *Tek-Cre:Eogt*$^{F/F}$, *Notch1*$^{+/-}$, and *Rbpj*$^{+/-}$ retinas. Arrows indicate fibrinogen staining outside vessels stained by IB4 (white). Three-dimensional images were constructed from confocal images by maximum intensity projection. (**B**) Higher magnification three-dimensional images of *Eogt*$^{-/-}$ retina constructed from confocal images using the Alpha-blend method. *Below*, single channel images showing fibrinogen (green) and IB4 (white) staining. (**C**) Sulfo-NHS-LC-biotin was perfused into P15 wild-type and *Eogt*$^{-/-}$ mice and extravasation determined immediately after perfusion by staining with CF488A-conjugated streptavidin (green) and Dylight594-conjugated IB4 (white). Three-dimensional reconstructions were created by maximum intensity projection. Enlarged images of boxed area are shown (right). (**D**) Sulfo-NHS-LC-biotin was perfused into P15 wild-type, *Eogt*$^{-/-}$, *Notch1*$^{+/-}$, *Eogt*$^{-/-}$*Notch1*$^{+/-}$ mice as in (**C**). Quantification of the number of extravasation sites in 210 × 210 μm squares (n = 6 per retina per mouse) is shown. Note that sulfo-NHS-LC-biotin extravasation in *Eogt*$^{-/-}$ retina is augmented in compound mutant mice. Data represent mean ± standard error; p values determined by Welch's t test. ***p≤0.001. (**E**) Whole-mount images of wild-type or *Eogt*$^{-/-}$ P15 retinas stained with IB4 (cyan) and anti-α SMA (magenta) antibody. (**F**) Whole-mount staining of wild-type and *Eogt*$^{-/-}$ P15 retinas using IB4 (white) together with anti-fibrinogen (green) and anti-NG2 (magenta) antibodies.

The following source data and figure supplement are available for figure 7:

*Figure 7 continued on next page*

*Figure 7 continued*

**Source data 1.** Raw data for *Figure 7D*.

**Figure supplement 1.** Sulfo-NHS-LC-biotin was perfused into P15 wild-type, *Eogt*$^{-/-}$, *Tek-Cre*, *Tek-Cre;Eogt*$^{F/F}$, *Notch1*$^{+/-}$, *Rbpj*$^{+/-}$, *Eogt*$^{-/-}$*Notch1*$^{+/-}$, *Eogt*$^{-/-}$*Rbpj*$^{+/-}$ mice and stained with CF488A-conjugated streptavidin (green) and Dylight594-conjugated IB4 (white) immediately after perfusion.

on vessel walls (*Figure 7A and B*), showing the leakage of plasma proteins. Similar to the blood-brain barrier, retinal vessels limit nonspecific transport between circulating blood and neural tissues of the retina (*Campbell et al., 2009*). Accordingly, perfused sulfo-NHS-LC-biotin remained within vessels in control P15 retinas, confirming an intact blood-retina barrier. In contrast, sulfo-NHS-LC-biotin was occasionally detected in extravascular spaces in perfused *Eogt*$^{-/-}$ retinas (*Figure 7C*). Taken together, these data suggested a partly impaired vascular integrity in the absence of *Eogt*.

Similar to *Eogt*$^{-/-}$ retinas, extravascular fibrinogen staining and extravasation of perfused sulfo-NHS-LC-biotin were observed in *Notch1*$^{+/-}$ or *Rbpj*$^{+/-}$ retinas (*Figure 7A* and *Figure 7—figure supplement 1*). Importantly, the combined loss of *Eogt* and a single *Notch1* allele in compound mutant *Eogt*$^{-/-}$*Notch1*$^{+/-}$ retinas gave a synergistic increase in extravasation of sulfo-NHS-LC-biotin along vessels (*Figure 7D*). The synergistic effect was also observed in *Eogt*$^{-/-}$*Rbpj*$^{+/-}$ retinas (*Figure 7D*). These results suggest that *Eogt* interacts with the Notch signaling pathway, thereby affecting retinal vascular integrity.

It has been reported that Notch activity is required for coverage of mural cells and vascular integrity (*Henshall et al., 2015*; *Liu et al., 2010*; *Wang et al., 2014*). In particular, lack of *Notch3* expression in mural cells results in decreased coverage of arteries with vascular smooth muscle cells (VSMC) (*Henshall et al., 2015*; *Liu et al., 2010*) and capillaries with pericytes (*Kofler et al., 2015*) in retinal vasculatures. Unlike *Notch3*$^{-/-}$ retinas, αSMA staining revealed intact coverage of retinal arteries with VSMC in P15 *Eogt*$^{-/-}$ mice (*Figure 7E*). Moreover, apparently normal pericyte investment was observed in P15 *Eogt*$^{-/-}$ retinas, regardless of the presence or absence of elevated fibrinogen staining (*Figure 7F*), suggesting that sufficient Notch activity is maintained in mural cells. In contrast, inactivation of *Eogt* in ECs in P15 *Tek-Cre:Eogt*$^{F/F}$ retinas showed aberrant fibrinogen staining and sulfo-NHS-LC-biotin extravasation (*Figure 7A* and *Figure 7—figure supplement 1*). Taken together, these results show that *Eogt* functions in ECs contribute to the integrity of retinal vessels by acting on Notch signaling activity.

## *Eogt* regulates Notch pathway target genes in ECs and ligand-induced Notch signaling

To investigate whether reduced EOGT directly impacts Notch target genes in ECs, brain ECs were isolated using anti-CD31 beads and gene expression was analyzed by qRT-PCR. *Eogt*$^{-/-}$ ECs showed reduced expression of Notch pathway targets (*Shawber et al., 2003*), including *Hes1* and *Hey1* (*Figure 8A*). Given that Notch signaling positively regulates *Dll4* expression (*Sacilotto et al., 2013*) (*Figure 8—figure supplement 1*), loss of EOGT would disrupt a positive feedback loop sustaining *Dll4* expression and Notch signaling. In fact, *Dll4* expression was markedly decreased in *Eogt*$^{-/-}$ ECs, whereas expression of *Jag1* and Notch receptors was maintained (*Figure 8A*). Previous reports showed that Notch signaling represses expression of vascular endothelial growth factor receptors (VEGFR) in ECs (*Ehling et al., 2013*; *Jakobsson et al., 2010*; *Suchting et al., 2007*; *Tammela et al., 2008*), which have crucial roles in the formation, function and maintenance of the vasculature (*Simons et al., 2016*). In *Eogt*$^{-/-}$ ECs, expression of *Vegfr2* and *Vegfr3*, but not *Vegfr1*, was slightly increased compared with wild-type ECs. In cerebral ECs, Notch and TGF-β signaling regulate the expression of N-cadherin, which is associated with maintaining vascular integrity (*Li et al., 2011*). Similarly, N-cadherin expression was relatively decreased in *Eogt*$^{-/-}$ ECs (*Figure 8A*).

To further investigate effects of EOGT on Notch target gene expression in ECs, DLL4-Fc or JAG1-Fc were coated on culture dishes and incubated with ECs derived from wild type or *Eogt*$^{-/-}$ lung. In unstimulated, control ECs, the expression level of Notch target genes including *Hey1* and *Hes1* was higher in wild type compared with *Eogt*-null cells (*Figure 8B*). Upon DLL4 or JAG1 stimulation, a further increase in *Hey1* and *Hes1* expression was observed in wild-type ECs (*Figure 8B*). In contrast, DLL4 failed to induce *Hey1* and *Hes1* expression in *Eogt*$^{-/-}$ ECs, whereas JAG1 induced a

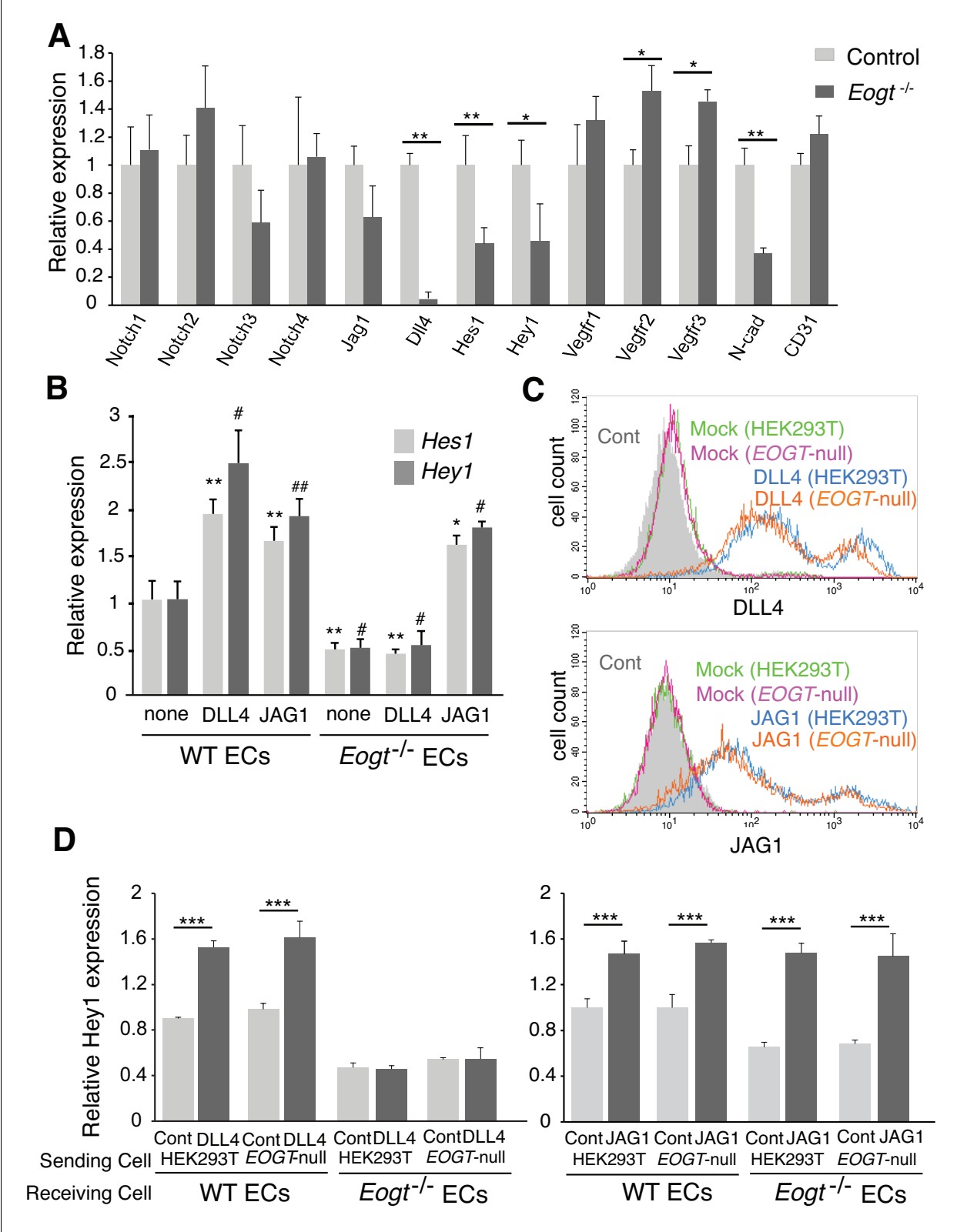

**Figure 8.** EOGT acts on Notch receptors to regulate ligand-induced Notch signaling in ECs. (**A**) Relative mRNA expression in purified brain EC cells. WT (*gray*) and *Eogt*$^{-/-}$ (dark gray) EC cells isolated from cerebrum using anti-CD31 antibody were analyzed for gene expression related to the Notch signaling pathway. Gene transcript levels were normalized to *Gapdh*. Data are mean ± S.D. from three independent experiments performed in triplicate. Each experiment analyzed pooled total RNA obtained from 10 mice. *p<0.05; **p<0.01 (Welch's t test). (**B**) qRT-PCR analysis of *Hes1* and *Figure 8 continued on next page*

*Figure 8 continued*

*Hey1* in wild type or *Eogt*$^{-/-}$ lung EC cells following stimulation with immobilized DLL4-Fc or JAG1-Fc. Gene transcripts were normalized to *Gapdh*. Data are mean ± S.D. from three independent experiments performed in triplicate. Each experiment analyzed total RNA obtained from a single mouse. Significance determined by Welch's t-test. *p<0.05; **p<0.01; #p<0.05; ##p<0.01 compared with the left-most *Hes*1 (*) or *Hey*1 (#) histogram. (C) Cells were transfected to express *Dll4* or *Jag1* and subjected to flow cytometry using DLL4 or JAG1 antibody, respectively. Comparison with mock-transfected cells indicated the amount of exogenously expressed DLL4 and JAG1 on the cell surface in wild-type or *EOGT*-null HEK293T cells. Mock-transfected HEK293T cells were labeled without primary antibody (*Cont.*). (D) *Dll4/Jag1*-transfected cells or Mock transfectants (signal-sending cells) were co-cultured with wild type or *Eogt*$^{-/-}$ lung EC cells (signal-receiving cells). qRT-PCR analysis of mouse *Hey1* expression suggested that EOGT is required for the ability to receive DLL4-mediated Notch signaling, and that EOGT is dispensable for DLL4 and JAG1 as inducers of Notch signaling. Gene transcript levels were normalized to *PECAM1* (CD31). Data are mean ± S.D. from three independent experiments performed in triplicate. Significance determined by Welch's t-test. ***p<0.001.

The following source data and figure supplement are available for figure 8:

**Source data 1.** Raw data for *Figure 8A,B,D*.

**Figure supplement 1.** *Dll4* expression is suppressed by inhibiting Notch signaling in ECs.

robust induction of these genes (*Figure 8B*). Taken together, these data suggest that EOGT acts as a positive regulator for DLL4-induced Notch signaling in EC cells.

## *Eogt* acts on Notch receptors to regulate ligand-induced Notch signaling

It has been reported that Notch ligands are also modified with O-glycans, which could affect Notch signaling activity (*Serth et al., 2015*; *Thakurdas et al., 2016*). To address the possibility that EOGT regulates Notch ligand function, HEK293T and *EOGT*-null cells were transfected to express JAG1 or DLL4 for co-culture with wild-type or *Eogt*$^{-/-}$ ECs. Exogenously expressed DLL4 and JAG1 in *EOGT*-null cells were expressed at the cell surface similarly to control HEK293T cells (*Figure 8C*). Thus, Notch ligands are presented on the cell surface in the absence of EOGT.

To determine if Notch ligands lacking O-GlcNAc stimulate Notch receptors, control and *EOGT*-null cells expressing DLL4 or JAG1 (Sending cells) were cultured with WT or *Eogt*-null ECs (Receiving cells), and the expression of Notch target gene *Hey1* was determined (*Figure 8D*). Both DLL4 and JAG1 lacking O-GlcNAc induced upregulation of *Hey1* in WT ECs similarly to controls. Thus, neither ligand needs O-GlcNAc to induce Notch signaling. *Hey1* expression was reduced in *Eogt*$^{-/-}$ ECs that also failed to respond to DLL4 sending cells. By contrast, co-culture of *Eogt*$^{-/-}$ ECs with JAG1 sending cells induced mouse *Hey1* expression to a level comparable with wild-type ECs (*Figure 8D*). Thus, Notch receptors in *Eogt*-null cells responded similarly to controls to JAG1 sending cells. These results suggest that EOGT is dispensable for DLL4 and JAG1 as inducers of Notch signaling, and that O-GlcNAc on Notch receptors is required for optimal DLL- but not JAG-induced Notch activation.

## Discussion

Here, we show that O-GlcNAc on Notch EGF repeats is a functional modification of Notch receptors that regulates Notch signaling and is required for optimal stimulation by DLL ligands. DLL4 constitutes the critical Notch ligand in regulating angiogenesis and vascular development (*Benedito et al., 2009*; *Hellström et al., 2007*). By contrast, JAG1 antagonizes Notch signaling during vasculogenesis (*Benedito et al., 2009*; *Pedrosa et al., 2015*). We show here that EOGT-catalyzed O-GlcNAc modification potentiates DLL1- and DLL4-mediated NOTCH1 signaling in co-culture assays and is required in vivo during retinal angiogenesis and the maintenance of retinal vascular integrity in mice. Site-directed mutagenesis of O-GlcNAc consensus sites in mouse NOTCH1 revealed that O-GlcNAc on EGF repeats located outside the canonical ligand-binding region affects DLL4-NOTCH1 binding. This O-GlcNAc in one or more EGF repeats may function directly in the binding of DLL4. Alternatively, O-GlcNAcylation may induce a conformational change of NOTCH1 that promotes DLL4 binding. Future structural analysis of O-GlcNAcylated NOTCH1 in complex with DLL4 will reveal how

O-GlcNAc potentiates DLL4-NOTCH1 interactions. Such a study revealed that DLL4 makes specific contacts with the O-fucose glycan in the ligand binding domain (EGF11 and 12) of NOTCH1 (*Luca et al., 2015*). O-fucose glycans on EGF12 affect both Delta and Jagged ligand-induced signaling as evidenced by the EGF12 O-fucosylation site mutant of NOTCH1 which has reduced signaling in response to DLL1 and JAG1 ligands (*Rampal et al., 2005*; *Shi et al., 2007*). An O-GlcNAc consensus site is present in EGF11, but not EGF12 in mammals, but the EGF11 site is not conserved among vertebrates.

The ability of EOGT to potentiate Delta ligands is similar to that of Fringe, a glycosyltransferase that transfers GlcNAc to O-fucose on Notch receptors (*Brückner et al., 2000*; *Moloney et al., 2000*). In mammals, three Fringe genes, *Mfng*, *Lfng*, and *Rfng*, are expressed in endothelial cells (*Benedito et al., 2009*). *Lfng*$^{-/-}$ mice exhibit enhanced sprouting and increased density of the vascular area in retinal angiogenesis (*Benedito et al., 2009*), a similarly mild phenotype to that in *Eogt*$^{-/-}$ retina. Thus, *Fringe* and *Eogt* might play complementary functions during retinal angiogenesis. However, *Mfng* inhibits JAG1-Notch signaling (*Benedito et al., 2009*), whereas we found no effect on JAG1-induced Notch signaling in assays using *Eogt*-deficient or *Eogt*-overexpressing cells. Thus, the distinct effects of the different GlcNAc modifications of Notch receptors indicate that EOGT and Fringe glycosyltransferases possess different physiological functions. Indeed, we observed no obvious abnormality in the skeleton (data not shown) of *Eogt*-null mice, making them phenotypically distinguishable from *Lfng*$^{-/-}$ mice (*Evrard et al., 1998*; *Zhang and Gridley, 1998*).

The phenotypic similarity and genetic interaction between *Eogt* and *Notch1* in retinas suggest that decreased Notch signaling caused by *Eogt* mutation is mediated by impaired O-GlcNAcylation of Notch1. However, we cannot formally exclude the possibility of involvement of other Notch receptors as EOGT substrates. In fact, all Notch receptors are potentially modified with O-GlcNAc and NOTCH2 was reported to be O-GlcNAcylated in mouse cerebrocortical tissue (*Alfaro et al., 2012*). Although roles for *Notch2* in vascular morphogenesis are not reported, *Notch1* and *Notch4* have partially overlapping roles (*Krebs et al., 2000*). However, a recent report revealed that DLL4 and JAG1 may fail to activate NOTCH4 (*James et al., 2014*). A detailed analysis will be required to address the contribution of other Notch receptors to EOGT-dependent regulation of Notch signaling.

The GlcNAc transferred by Fringe or EOGT can be extended with galactose (*Chen et al., 2001*; *Sakaidani et al., 2012*). Interestingly, inactivation of galactosyltransferase *B4galt1* in Lec20 CHO cells abrogated the effects of LFNG on Notch signaling (*Hou et al., 2012*). Consistent with roles for galactose in Notch regulation, embryos lacking *B4galt1* exhibit subtle Notch signaling defects during somitogenesis (*Chen et al., 2006*). It would be of interest to investigate roles for galactose in retinal angiogenesis and vascular integrity.

Our prediction of O-GlcNAc sites on Notch receptors relied on the consensus sequence $C^5XXG$ $(Y/F/L)(T/S)GX_{2-3}C^6$ derived from previously reported O-GlcNAcylated sequences (*Alfaro et al., 2012*; *Sakaidani et al., 2011*). In the course of preparation of this manuscript, a study in *Drosophila* Notch revealed that a Y/F/L residue at the $-1$ site is not absolutely required for the modification (*Harvey et al., 2016*). Therefore, $C^5XXGX(T/S)GX_{2-3}C^6$ can be proposed as a broad consensus site. Nonetheless, removing only four modification sites (EGF2, 10, 17, 20) out of 11 $C^5XXG(Y/F)(T/S)$ $GXXC^6$ sequences drastically decreased the O-GlcNAc level on NOTCH1 (*Figure 2*). These results suggest that preferred O-GlcNAcylation requires additional constraints. Consistent with this view, five domains (EGF4, 11, 12, 14 and 20) out of 15 *Drosophila* Notch EGF repeats containing the $C^5XXG(Y/F)(T/S)GXXC^6$ consensus were found to be major O-GlcNAcylation sites in embryos (*Harvey et al., 2016*) (*Figure 2—figure supplement 1*). It is also noteworthy that *Drosophila Eogt* mutants fail to display an obvious loss of Notch signaling phenotypes (*Müller et al., 2013*; *Sakaidani et al., 2011*). Nevertheless, Notch pathway mutations were found to suppress the wing blistering phenotype in *Eogt* knockdown flies (*Müller et al., 2013*). Thus, *Eogt* may affect Notch signaling in a tissue-specific manner. However, O-GlcNAc consensus sites on EGF repeats are poorly conserved between *Drosophila* Notch and mammalian NOTCH1. Moreover, the elongation of O-GlcNAc by galactose is not observed in *Drosophila* embryos (*Harvey et al., 2016*). It would be of interest to determine whether the O-GlcNAc Notch-ligand interaction we report here is conserved in *Drosophila*.

The combined data on roles for O-glycans in Notch signaling suggest that each of the responsible glycosyltransferases may have different effects on Notch receptor functions. Thus, POFUT1 (Ofut1 in

*Drosophila*) and POGLUT1 (Rumi in *Drosophila*), which catalyze O-fucosylation and O-glucosylation, respectively, regulate the strength of Notch activation. Genetic inactivation of *Pofut1/ofut1* in *Drosophila* results in diminished ligand binding to Notch and *Poglut1/rumi* mutation results in impaired Notch processing (*Haltom and Jafar-Nejad, 2015*; *Stanley and Okajima, 2010*). Subsequent additions to O-fucose and O-glucose regulate subcellular localization or change the affinity toward Notch ligands. In *Drosophila*, xylose on O-glucose negatively regulates Notch signaling by reducing cell surface expression of Notch (*Lee et al., 2013*). The distinct effects of O-glycans on Notch receptors, coupled with differential expression of the relevant glycosyltransferases explain why different human diseases arise from mutation of *POFUT1* (Dowling-Degos disease), *LFNG* (Spondylocostal Dysostosis) and *EOGT* (AOS) (*Basmanav et al., 2014*). Unexpectedly, *Eogt*-null mice do not exhibit abnormalities predicted from the symptoms of AOS patients. This was also the case in *Eogt*-null mice homozygous for *Eogt* gene disruption by a neo cassette (H. Yagi and K. Kato, personal communication). However, the vascular defects observed in $Eogt^{-/-}$ mice are consistent with the view that small-vessel vasculopathy may be a basis for pathologies in AOS.

## Materials and methods

### Antibodies

Antibodies used in microscopy, flow cytometry and Western blot experiments: biotinylated isolectin B4 (IB4; Vector [B-1105]), Cy3-conjugated anti–α-smooth muscle actin (αSMA) antibody (clone 1A4; Sigma [C6198] or fluorescein isothiocyanate(FITC)-conjugated anti-αSMA antibody (clone 1A4; Sigma [F3777]), rabbit anti-human EOGT antibody (Sigma [HPA019460]), mouse anti-O-GlcNAc antibody (CTD110.6; Thermo Scientific [24565] or Sigma [07764]) (*Comer et al., 2001*), hamster anti-mouse NOTCH1 ECD antibody (8G10, Santa Cruz [sc-32756]), sheep anti-hamster NOTCH1 ECD antibody (R and D Systems, AF5267), rabbit anti-human NOTCH1 ECD antibody (H-131, Santa Cruz [sc-9170]), rabbit anti-human NOTCH1 ICD antibody (D6F11, Cell signaling), rabbit anti-mouse activated NOTCH1 (Val1744, Cell Signaling Technology [4147]) (*Huppert et al., 2000*), sheep anti-BiP antibody (BD Biosciences [51–9001980]), rabbit anti-NG2 chondroitin sulfate antibody (Millipore [AB5320]), rat anti-NG2 chondroitin sulfate antibody (R and D Systems [MAB6689]), mouse anti-Myc antibody (4A6, Upstate [05-724]), hamster anti-mouse JAG1 antibody (HMJ1-29; Biolegend [130902]), goat anti-mouse DLL4 antibody (R and D Systems [AF1389]), rabbit anti-fibrinogen antibody (Dako [A0080]), FITC-conjugated anti-hamster IgG antibody (Cappel [55400]), CF488A-conjugated anti-goat IgG antibody (Sigma [SAB4600032]), Dylight488–conjugated anti-rabbit IgG (Vector Labs [DI-1488]), CF488A-conjugated streptavidin (Biotium [29034]), CF640R-conjugated anti-rat IgG (Sigma [SAB4600156]), Rhodamine Red-X-conjugated donkey anti-sheep IgG (Jackson ImmunoResearch), Dylight649-conjugated streptavidin (Vector Labs [SA5649]), horseradish peroxidase (HRP)-conjugated goat anti-mouse IgM (Thermo Scientific [31444]), HRP-conjugated goat anti-rabbit IgG (Invitrogen [65–6120]), HRP-conjugated goat anti-Armenian hamster IgG (Santa Cruz [sc-2443]), HRP-conjugated horse anti-mouse IgG antibody (Cell signaling [7076S]), alkaline phosphatase (AP)-conjugated anti-FITC antibody (Roche [11-426-338-910]), FITC-conjugated IB4 (Vector Labs [FL-1201]), and Dylight594-conjugated IB4 (Vector Labs [DL-1178]). A new antibody was raised in rabbit against mouse EOGT lacking the first 19 amino acids (aa 20–527), produced in *E. coli*. For the purification of IgG, the rabbit serum was subjected to affinity chromatography on Protein A–Sepharose CL-4B (GE Healthcare Life Science).

### Plasmids

A mouse *Notch1* expression plasmid (pTracer-CMV/*Notch1*) encoding full-length native Notch1 (nucleotide numbers from −82 to 9378) was obtained from S. Chiba. For generating in situ hybridization probes, full length mouse *Eogt* cDNA created by PCR was cloned into pBluscript II SK(-) vector. For transfection of *Eogt,* the cDNA was cloned into pSectag2-IRES-GFP and pEF1 vectors (*Sakaidani et al., 2012*). A human *EOGT* cDNA in pCR3.1 was previously described (*Tashima and Stanley, 2014*). *Notch1* with O-GlcNAc site mutations were generated by the Site-Directed Mutagenesis Kit (Stratagene [200518]) to introduce the following mutations: 280T→G (S94A in EGF2); 1213A→G (T405A in EGF10); 2011A→G (T671A in EGF17); 2350AG→GC (S784A in EGF20). An

expression vector for *Dll4* (pSport6-DLL4) and *Jag1* (pBOB-Jag1) were obtained from DNAFORM (Clone ID: 4017786) or provided by Gerry Weinmaster, respectively.

## Cell culture and transfection

HEK293T cells were kindly provided by Professor Tsukasa Matsuda (Nagoya University) and maintained in Dulbecco's modified Eagle's medium (DMEM) supplemented with 7.5% fetal bovine serum (FBS). Transfection into HEK293T cells was performed using polyethylenimine, MW 4000 (Polysciences [24885]). In brief, 4 μg of plasmid DNA was diluted in 200 μl of Opti-MEM I medium (Gibco [31985]), which was then mixed with 12 μg of polyethylenimine and incubated for 30 min. Prior to transfection, HEK293T cells cultured in 6-well dish were pretreated with 800 μl of Opti-MEM I medium and supplemented with the DNA/ polyethylenimine mixture. After 4 hr of incubation, the medium was replaced with 2 ml of 7.5% FBS in DMEM containing penicillin/streptomycin followed by incubation for 48 hr before analysis. HeLa and CHO Lec1 cells were maintained in α-MEM containing 10% FBS. Transfection into HeLa and CHO Lec1 cells was performed using X-treme Gene 9 (Roche [06365779001]) as described by the manufacturer. Absence of mycoplasma contamination was confirmed upon receipt of the cell lines, and all the experiments were completed within 6 months from the contamination test.

## Generation of knockout cells using CRISPR/Cas9 and knockdown cells using shRNA

### EOGT targeted by CRISPR/Cas9

HEK293T cells were transiently transfected with GFP-Cas9 and guide RNA (gRNA) expression plasmid (Sigma), encoding a single gRNA (5'-GCCAGCATCCGCTTGCCAGAGG-3') targeting the fourth coding exon of *EOGT*. Single Cas9-GFP-positive cells were obtained by limiting dilution in 96-well plates. GFP-positive clones were expanded and analyzed for deletion of the targeted genomic region by PCR (forward primer: 5'-CCATTTGCTAACGTGGTGCC-3', reverse primer: 5'-GCAGAACC TGTAAGCCACCT-3') and sequencing. Absence of EOGT was demonstrated by immunoblotting with CTD110.6 O-GlcNAc antibody and our new rabbit Eogt antibody.

### Notch1 targeted by CRISPR/Cas9

HEK293T cells were transiently transfected with GFP-Cas9 together with gRNA expression plasmid encoding a single gRNA (5'-GGTTGGGGTCCTGGCATCGCGG-3'), which targets the third exon of *NOTCH1*. Individual clones were screened for deletion of the target sequence by PCR (forward primer: 5'-CCGCGATGCCAGGACCCCAACC-3', reverse primer: 5'- GTCAGGCAGAGGGGCCCA-GAGA-3'). Genomic sequence at Notch locus was analyzed by sequencing of a PCR-amplified fragment (forward primer: 5'-GGCATGTGTCTGGTTCCAGA-3', reverse primer: 5'-CCCGCCAAGTACC TCAAGTT-3').

### EOGT targeted by shRNA or siRNA

Four shRNA constructs targeting human *EOGT* were prepared in Lentivirus for transduction of HeLa cells. TA197 and TA198 shRNAs in pGIPZ were from the GIPZ Lentiviral Human shRNAmir library (Open Biosystems, ThermoScientific), and TA195 and TA196 shRNAs in pLKO.1 were from the TRC Lentiviral Human Genome shRNA library (RNAi Consortium). The *EOGT* sequences target the coding region (TA196 and TA198) or the 3'UTR (TA195 and TA197) as follows: TA195: AGAGTGTTCCAAC TTGTTT; TA196: AGACCTATAGAAGATGCTA; TA197: GCATGAAAGTATTCCAGTCTT; TA198: GACAGATTTAAGGAGGACTTT. HEK293T cells on a 10 cm dish were transfected with 10 μg shRNA plasmid, 3 μg ENV plasmid (VSV-G), 5 μg pMDLg/pRRE (Packaging plasmid III Gen. Pack) and 2.5 μg REV plasmid combined in a calcium phosphate precipitate as described (*Follenzi and Naldini, 2002*). Plasmids were obtained from Ana Maria Cuervo, Albert Einstein College Medicine. After incubation at 37°C in 5% $CO_2$ for 12–14 hr, DMEM medium (Life Technologies [11965–092]) containing 10% heat-inactivated fetal calf serum (FCS; Gemini [100-800]) was added. After 36 hr, medium containing Lentivirus was collected, centrifuged at 2000 rpm for 10 min, filtered through a 0.45 μm filter, and frozen at −80°C in 1 ml aliquots. For transduction, HeLa cells on a 60 mm dish were incubated with 1 ml virus diluted with 5 ml RPMI medium (GIBCO [11875–093]) containing 10% FCS, 8 μg/ml Polybrene (Sigma [10768–9]) and penicillin and streptomycin (GIBCO [15140–122]). After 2 hr

at 37°C, medium was removed and fresh Lentivirus in Polybrene was added. After 2 hr at 37°C, the medium was replaced with fresh RPMI, 10% FCS with antibiotics, and incubation continued for 44 hr. To select stable cell populations, the transduced HeLa cells were trypsinized and replated in 1–2 µg/ml puromycin (Sigma [P8833]). Puromycin-resistant cell populations were expanded and stored frozen in medium containing 10% FCS and 10% dimethylsulfoxide (DMSO; Fisher Scientific [D128-500]) at −135°C. Three independent preparations of HeLa cells expressing the *GAPDH* shRNA, or each of the four *EOGT*-targeting shRNAs, were used in experiments. Lec1 CHO cells expressing siRNAs targeting hamster *Eogt* were previously described (*Tashima and Stanley, 2014*). *Eogt* transcript levels in Lec1 and HeLa cells were determined by qRT-PCR using the Super Script First Strand synthesis kit (Invitrogen [11904–018]) for reverse transcription and SYBR green (Thermo Scientific [AB-1453]) for real time PCR, following the manufacturer's instructions.

## Notch ligand binding assays

### Flow cytometry assay

Cells were washed with ligand binding buffer (Hanks' buffered salt solution, pH 7.4, 1 mM $CaCl_2$, 1% (w/v) BSA (Gemini [700–100P]), 0.05% $NaN_3$), fixed in 4% PFA in PBS-CMF at RT for 10 min and stored in ligand binding buffer at 4°C. Fixed cells ($2 \times 10^5$) were incubated with 50 µl ligand binding buffer containing 100-750 ng/ml DLL1-Fc, DLL4-Fc or JAG1-Fc at 4°C for 1 hr. Soluble ligands were prepared as described previously (*Stahl et al., 2008*; *Song et al., 2016*). After incubation, cells were washed twice with 0.5 ml ligand binding buffer and incubated with phycoerythrin (PE)-conjugated goat anti-human IgG (Fc-specific) antibody (Jackson ImmunoResearch [109-116-170], 1:100 in ligand binding buffer) at 4°C for 30 min in the dark. The cells were then washed twice with 1 ml of ligand binding buffer and analyzed in a FACSCalibur flow cytometer. For detection of NOTCH1 at the cell surface, fixed cells were incubated with sheep anti-mouse NOTCH1 antibody AF5267 (1:50 in ligand binding buffer) at 4°C for 1 hr, washed and incubated with Rhodamine Red-X-conjugated donkey anti-sheep IgG (1:100 in ligand binding buffer) at 4°C for 30 min in the dark. Cells were washed twice with 1 ml of ligand binding buffer, and analyzed using a FACSCalibur flow cytometer. The Notch ligand binding flow cytometry assay is described in detail in Bio-protocol (*Varshney and Stanley, 2017*).

### Dynabead assay

A *Notch1* expression vector (pTracer-CMV/*Notch1*) was transfected alone or together with a vector expressing mouse *Eogt* (pSecTag2/Hygro/*Eogt*) into wild type or *EOGT*-null HEK293T cells. The detailed protocol is available in Bio-protocol (*Sawaguchi et al., 2017*). To identify transfected cells, a GFP-expression vector (pMX-GFP) was included at 1/8 the total DNA transfected. After 48 hr, 20 µl Protein A Dynabeads (Veritas [10002D]) pre-coated with 500 ng of DLL4-Fc (Invitrogen [10171 H02H]) or JAG1-Fc (Invitrogen [11648 H02H]) were added to the culture media, and incubated for 2 hr at 37°C. Transfected cells were rinsed 10 time with PBS, and fixed with 4% PFA in PBS for 20 min at RT. The number of beads bound per 50 GFP-positive cells was counted for quantification.

## Immunofluroescence microscopy

HEK293T cells were fixed with 4% paraformaldehyde in PBS for 20 min and ice-cold methanol for 5 min. After washing with PBS, cells were blocked with 5% FBS in PBS for 30 min. Immunostaining was performed overnight at 4°C by incubation with anti-O-GlcNAc mAb CTD110.6 (1:1000) and anti-NOTCH1 ECD Ab 8G10 (1:200) diluted in 5% FBS/PBS. After washing with cold PBS, cells were incubated with Alexa Fluor 555-conjugated goat anti-Mouse IgM heavy chain antibody (2 µg/ml; Invitrogen [A21426]) and DyLight 488-conjugated goat anti-rabbit IgG antibody (3 µg/ml; Vector Labs [DI-1488]) diluted in 5% FBS/PBS for 2 hr at RT, washed with cold PBS. Cells were mounted on slides using DAPI-Fluoromount-G (Southern Biotech, [0100–20]) for observation with a FSX 100 fluorescent microscope (Olympus).

## Flow cytometry

Cultured HEK293T cells were detached by pipetting 10 times. After washing with cold PBS, cells were incubated with hamster anti-Notch1 (8G10; 4 µg/ml) antibody, goat anti-DLL4 (0.1 µg/ml), or hamster anti-JAG1 (0.5 µg/ml) antibody diluted in 5% FBS in PBS on ice for 30 min. After washing

with cold PBS, cells were incubated with FITC-labeled anti-hamster IgG (40 µg/ml) or FITC-labeled anti-goat IgG (10 µg/ml) antibody in 5% FBS in PBS on ice for 30 min, washed with cold PBS, and subjected to flow cytometry using a FACSCalibur cytometer (Becton Dickinson).

## Immunoprecipitation

For immunoprecipitation of mouse NOTCH1 from transfected HEK293T cells, cells were lysed in Cell Lysis Buffer (Cell signaling [9803S]). Cell lysates were incubated with 8G10 Notch1 antibody at the concentration of 2 µg/ml for 2 hr at 4°C, and incubated with Protein G Sepharose (GE Healthcare [17-0618-01]) for 2 hr at 4°C. After extensive wash with 50 mM Tris-HCl, pH 7.4, 150 mM NaCl, and 0.25% Triton X-100, NOTCH1 was eluted with SDS-PAGE sample buffer containing 2% SDS and 70 mM 2-mercaptoethanol. Immunoblotting for the detection of O-GlcNAc on NOTCH1 was performed using CTD110.6 O-GlcNAc (1:2000) or 8G10 NOTCH1 (1:500) antibodies, followed by HRP-conjugated anti-mouse IgM (1:4000) or anti-hamster IgG antibodies (1:1000), respectively, and enhanced chemiluminescence as described previously (*Ogawa et al., 2015*).

For purification of NOTCH1 EGF repeats (*Notch1*-EGF1-36-MycHis) from cell lysates, cells were transfected with pSegtag2/Hygro/*Notch1*-EGF:mycHis (*Hou et al., 2012*) alone or together with pSecTag2/Hygro/*Eogt*. Cell lysates were added with 2 µg of anti-Myc antibody (4A6). After incubation for 2 hr at 4°C, Protein G Sepharose was added to the lysates, which were then incubated for 2 hr at 4°C. Immunoprecipitates were washed and eluted with SDS-PAGE sample buffer as described above. Immunodetection was performed using anti-Myc antibody (1:1000) or CTD110.6 (1:2000) antibodies, followed by HRP-conjugated anti-mouse IgM (1:4000) or anti-mouse IgG antibodies (1:2000), respectively, and enhanced chemiluminescence.

## Ligand-induced Notch signaling assays

### NOTCH1 activation assay

HeLa cells stably expressing Lentivirus with an shRNA targeted against *EOGT* or *GAPDH* were plated in 6-well plates at $2 \times 10^5$ cells/well in RPMI containing 10% FCS. After 24 hr, the medium was replaced with 2 ml medium containing either DMSO (vehicle) or gamma-secretase inhibitor (GSI) in DMSO for 1 hr. The GSIs L-685,458 (Sigma [L1790]) or N-[N-(3,5-difluorophenacetyl)-L-alanyl]-S-phenylglycine t-butyl ester (DAPT; Sigma [D5942]) were used at 1–2 µM for HeLa. Co-culture was initiated by overlaying puromycin-resistant HeLa cells with $2.4 \times 10^6$ L cells, or L cells expressing DLL1 (D1/L) in medium containing vehicle or GSI, as described previously (*Lu et al., 2010*). After 6–8 hr at 37°C, less adherent L cells were partially removed by washing twice with 2 ml PBS containing 1 mM CaCl$_2$ and 1 mM MgCl$_2$, pH 7.2. Remaining cells were lysed in 0.5 ml RIPA buffer (Millipore [20-188]) containing 0.1% SDS and Complete mini, EDTA-free protease inhibitors (Roche [11836170001]), and scraped into a microfuge tube. After centrifugation at 15,000g for 15 min at 4°C, the supernatant was added to 120 µl glycerol, mixed and frozen at −80°C in 100 µl aliquots. For SDS-PAGE, SDS gel loading buffer with fresh dithiothreitol (DTT, 30 mM) was added, and the samples heated at 80°C for 15 min prior to separation on a 7.5% SDS-PAGE gel. Proteins were transferred to polyvinylidene fluoride (PVDF) membranes in transfer buffer containing 10% methanol. Membranes were blocked in 5% non-fat dry milk in 0.05% Tween-20% and 1% Thimerosal (Sigma [T5125]), and washed twice in TBS/Tween (10 mM Tris HCl, 0.15 M NaCl, 0.05% Tween 20). Membranes were cut into three sections and incubated in primary antibodies to detect NOTCH1 full length (~280 kDa), activated, cleaved NOTCH1 (~110 kDa) and EOGT (~52 kDa). Antibody against NOTCH1 ECD (AF5267) was used at 1:250 in TBS containing 3% cold fish gelatin, 1% BSA and 1% Thimerosal and incubated overnight at 4°C. After four washes in TBS/Tween, the membrane was incubated in rabbit anti-sheep IgG conjugated to HRP diluted 1:5000 in blocking solution. All buffer reagents were from Sigma. Antibodies to detect activated NOTCH1 (Val1744) and EOGT were incubated in the same manner at ~1:1000 dilution. Secondary Ab was 1:2000–10,000 goat anti-rabbit IgG-HRP. Detection of O-GlcNAc was performed after stripping NOTCH1 mAb using mAb CTD110.6 diluted ~1:1000–1:5000, incubated as above and detected by 1:5000 goat anti-mouse IgM-HRP. After four washes in TBS/Tween, membranes were treated with ECL reagent from Pierce (West-Pico [34095] or West-Femto [34080]) for 5 min, the membrane sections reassembled, and exposed for various times to film (Denville Scientific [E3018]). Stripping for reprobing membranes was performed as described (*Yeung and Stanley, 2009*).

## Notch reporter assay

To assay ligand-induced stimulation of the Notch reporter gene TP1-luciferase, co-culture assays using Lec1 CHO cells stably expressing siRNA targeted against *Eogt* (*Tashima and Stanley, 2014*) overlaid with L cells or L cells expressing DLL1, were performed as described previously (*Lu et al., 2010*; *Stahl et al., 2008*). TP1-luciferase activity was normalized against co-transfected Renilla luciferase. Cultures were performed in duplicate with and without 12.5 μM GSI IX (Calbiochem [565784]) in DMSO. Rescue of *Eogt* knockdown was performed by inclusion of a human *EOGT* cDNA in the transfection mix.

## Notch target gene assay with solid-phase ligand

To assay ligand-induced activation of Notch target genes, each well of a 96-well MicroWell plate (Thermo Scientific, [442404]) was coated with 500 ng of DLL4-Fc or JAG1-Fc in PBS and incubated for 16–18 hr at 4°C. Lung ECs ($1 \times 10^5$) from wild-type or $Eogt^{-/-}$ mice were plated into each well in 200 μl complete HuMedia-EG2. After 16 hr incubation at 37°C in a 5% $CO_2$ incubator, total RNA was isolated from the stimulated ECs using TRIzol Reagent (Ambion, [15596018]) for qRT-PCR analysis.

## Notch target gene assay by co-culture

Wild type or *EOGT*-null HEK293T cells were transiently transfected with pSport6-*Dll4* or pBOB-*Jag1* plasmid. After 24 hr, $1.0 \times 10^5$ ligand-expressing cells were added onto wild type or $Eogt^{-/-}$ lung ECs plated in 96-well dish and incubated for 24 hr at 37°C in a 5% $CO_2$ incubator. Total RNA was isolated using PureLink RNA Mini Kit (Ambion, [12183020]) for qRT-PCR analysis.

### *Eogt* targeting vector

BAC clone RP24-388B15 derived from C57BL/6J mice and containing a genomic region that encodes *Eogt* was obtained from the BACPAC Resource Center. Conditional knockout (cKO) targeting vector (pDT-*lox*P-*lox*P-FRT-PGKneo-*lox*PFRT) was obtained from Satoru Takahashi (Tsukuba University).

Mouse genomic DNA fragments including *Eogt* exon 10, a 7.8 kb upstream region (long arm), or 1.9 kb downstream region (short arm) were amplified by PCR with PrimeSTAR Max polymerase (Takara [R045A]). Each fragment was cloned into the *Pml*I site, *Pme*I/*Kpn*I sites, or the *Xho*I site of the cKO vector, respectively, using the In-Fusion Advantage PCR Cloning Kit (Clontech [639619]). All insert sequences were confirmed by DNA sequencing.

*Eogt*-targeting vector DNA was linearized with AscI, and introduced into C57BL/6J ES cells (*Tanimoto et al., 2008*) by electroporation. Genomic DNA isolated from G418-resistant clones was subjected to PCR screening to confirm homologous recombination using KOD FX polymerase (Toyobo [KFX-101]), followed by a second screening to confirm the position of the third *lox*P sequence using LA Taq polymerase (Takara [RR002A]). Selected ES clones were subjected to Southern blotting using long-arm (900 bp) or short-arm (1 kb) probes radiolabelled with [$^{32}$P]dCTP using the Megaprime Kit (GE Healthcare [RPN1606]). Genomic DNA was digested with *Kpn*I/*Sal*I and hybridized with the long-arm probe, or digested with *Hind*III and hybridized with the short-arm probe. Two clones were microinjected into blastocysts that were implanted into pseudopregnant female mice in the Tsukuba University transgenic core facility using standard methodologies. Primers used for PCR are shown in *Table 1*.

### Mice

*Eogt*$^{floxed/neo}$ mice were crossed with *Ayu1-Cre* (*Niwa et al., 1993*) or *FLPe* mice (B6;SJL-Tg (ACTFLPe)9205Dym/J backcrossed more than nine times to C57BL/6J) to generate mice carrying floxed or exon 10-deleted *Eogt* alleles, respectively, in collaboration with Satoru Takahashi (*Moriguchi et al., 2006*). Mice with a floxed *Rbpj* allele or a *Tek-Cre* transgene were obtained from T. Honjo (*Han et al., 2002*) and M. Yanagisawa (*Kisanuki et al., 2001*), respectively, and provided by the RIKEN BRC through the National Bio-Resource Project of the MEXT, Japan. *Rbpj-null* mice were generated by mating *Rbpj*$^{F/F}$ mice with *Ayu1-Cre* deleter mice. *Tek-Cre: Eogt*$^{F/F}$ mice were generated in collaboration with Kenji Uchimura (Nagoya University). *Notch1* null mice (Notch1$^{tm1-Con}$/J) (*Conlon et al., 1995*) were obtained from the Jackson laboratory. Mice with a ligand binding domain mutation in *Notch1* (*Notch1*$^{lbd}$) and mice lacking the O-fucose site in EGF12 of *Notch1*

**Table 1.** Primers used for cloning DNA fragments, screening ES cells, or genotyping by PCR.

Target region
Exon10 Fw 5'–<u>ATACGAAGTTATCACC</u>GAACCTAGCCCATATTT–3'
Exon10 Rv2 5'–<u>ACGAAGTTATGTCGA</u>CGACTGAGCATTGCTGTT–3'

Long arm region
Long arm Fw1 5'–<u>CGAATCAAGCTGTTT</u>GGTCCATTCTCTGCTCCA–3'
Long arm Rv 5'–<u>ACGAAGTTATGGTAC</u>GGTCAACTTGAAGAAGTA–3'

Short arm region
Short arm Fw 5'–<u>TAGGAACTTCCTCGA</u>AATTCAGTGCTTAGAAGT–3'
Short arm Rv1 5'–<u>GCGCGCCTTTCTCGA</u>ACACTGTGTACAGTGACA–3'

Long-arm probe
larm16380Fw 5'–CTGCCTCAGCTTCCTGAGTG–3'
larm17196Rv 5'–CATGTCAGATCAGACAGTTC–3'

Short-arm probe
sarm26294Fw 5'–CTGAGCTATGTACTGGATGC–3'
AscI-sarmRv2 5'–TGAAGAGGCGCGCCCAGAGACAGAAAAGCAC–3'

ES cells first screening
PGK S1 5'–CCTCCCCTACCCGGTAGAATTGACC–3'
GL1 typing RV2 5'–GAACTGTCAGATTTGGTGACACAGAAAGGC–3'

ES cells second screening
3rdlox Fw 5'–CCACCCGACCCCTGCCAGAACATAATGCTCTCTTGCATC–3'
3rdlox Rv 5'–GCTGTCGCCAGAGGAGAGAGTGGGTGCTTACTTAC–3'

*Eogt* mice genotyping
3rdloxFw 5'–CCACCCGACCCCTGCCAGAACATAATGCTCTCTTGCATC–3'
3rdloxRv2 5'–GCTGTCGCCAGAGGAGAGAGTGGGTGCTTACTTAC–3'
25307Rv 5'–CCAAGGCGGTCTTGGCCCAT–3'

RT-PCR for *Eogt*
Exon 9 Fw 5'– AGGCTACACGCAGCTCAATT –3'
Exon 11 Rv 5'– AGAAGCCGTGTTTTCGTTGC –3'

qRT-PCR for *Eogt*
Exon 1 Fw 5'–AAGCTGCAGGTCCGTGAAAA–3'
Exon 2 Rv 5'–TAGGTTAGGCTACCGCGTCT–3'

Underlined sequences are 15 bp homologous overlaps required for In-Fusion cloning.

(*Notch1*[12f]) were previously described (**Ge and Stanley, 2008**, **2010**). *Notch1*[12f] mice were used after backcrossing >10 times to C57Bl/6. Backcrossed *Notch1*[12f/12f] progeny die around mid-gestation (manuscript in preparation).

All experimental procedures were performed in accordance with the Guidelines for Animal Experimentation in Nagoya University Graduate School of Medicine and Japanese Government Animal Protection and Management Law (Permit Number: 26397). Animal experiments at the Albert Einstein College of Medicine were performed with the approval of the Institutional Animal Care and Use Committee (Permit Number: 20140803).

## Whole mount staining of retina

Whole mount staining of mouse retina was performed as reported previously (**Tual-Chalot et al., 2013**), unless otherwise noted. Briefly, eyes were fixed with 4% paraformaldehyde (PFA) in phosphate buffered saline without divalent cations (PBS-CMF), pH 7.4 at room temperature (RT) for 15 min, except for preserving filopodia at the vascular front, eyes were fixed in 4% PFA in PBS-CMF at 4°C for 2 hr. After dissection of retinas in PBS-CMF, flat retinas were prepared by dropping cold methanol. After washing with PBS-CMF, retinas were incubated with Perm/Block solution containing PBS-CMF, pH 7.4, 0.3% TritonX-100, and 0.2% bovine serum albumin (BSA) supplemented with 5% donkey or goat serum for 1 hr at RT. Immunostaining was performed overnight at 4°C by incubation with biotin-IB4 (1:100 or 1:250) and Cy3- or FITC-conjugated anti-αSMA diluted in Perm/Block solution (1:200), followed by four washes with PBS-CMF containing 0.3% TritonX-100 (PBSTX) for 10 min at RT, followed by incubation overnight at 4°C in SA-488 (1:200) or Dylight649-conjugated streptavidin (1:250) in Perm/Block solution. Alternatively, retinas were directly labeled with Dylight594-conjugated IB4 (2.5 μg/ml in Perm/Block). After four washes with PBSTX for 15 min at RT and a rinse with

PBS-CMF, retinas were mounted using prolong diamond anti-fade mounting medium (Life Sciences, [1664835]) for observation using a TiE-A1R-KT5 microscope (Nikon) or a P250 High Capacity Slide Scanner (Perkin Elmer [1S10OD019961-01]). Whole mount staining for NG2+ pericytes was performed following the procedure described above using rabbit anti-NG2 antibody (1:200) or rat anti-NG2 antibody (1:1000), and DyLight488-conjugated anti-rabbit IgG (1:200) or CF640R-conjugated anti-rat IgG (1:200) as primary and secondary antibodies, respectively. Whole mount staining of extravasated fibrinogen was performed as described above using rabbit anti-fibrinogen antibody (1:200) as a primary antibody and DyLight488-conjugated anti-rabbit IgG (1:200) as a secondary antibody.

## Quantification of retinal angiogenesis

The length of $\alpha$SMA$^+$ blood vessels was calculated by determining the average of the 3–4 longest $\alpha$SMA$^+$ vessels per retina. The number of vascular branch points was measured by analyzing 3–8 fields of 500 × 500 μm per retina which were chosen randomly to cover the region under the vascular front for P5, or central regions for P15 retinas. AngioTool (*Zudaire et al., 2011*) was used to calculate branch points. For counting the number of filopodia, 4–12 random fields of 250 × 250 μm per retina at the vascular front, and the number of filopodia were counted by eye and marked with a red dot in NIH ImageJ (*Schneider et al., 2012*).

## In situ hybridization

Eyes from newborn mice were fixed with 4% PFA/PBS for 15 min at RT. Retinas were dissected in 4% PFA in PBS. Flat retinas were prepared by dropping cold methanol onto the retina and stored in methanol at −20°C. In situ hybridization was performed as described previously (*Powner et al., 2012*) with modifications. In brief, flat retinas were washed with PBS, incubated with 1 μl of FITC-labeled probe in diluted in 1 ml of hybridization buffer overnight at 65°C, washed with formamide wash buffer for 15 min 3 times at 65°C, followed by PBS containing 0.1% Tween-20 for 20 min, twice at RT. Retinas were incubated in 500 μl IHC blocking buffer (3% Triton X-100, 0.5% Tween-20, 1% FBS in 2x PBS) for 20 min at RT, followed by AP-conjugated anti-FITC antibody (1:200 dilution) in IHC blocking buffer. Counter staining was performed by incubating retinas in Dylight 594-conjugated IB4 (2.5 μg/ml) in IHC blocking buffer. After incubation overnight at 4°C on a slow shaker, retinas were washed with PBS containing 0.1% Tween-20 (PTW) for 20 min twice at RT, and incubated with BCIP/NBT solution (Roche [11681460001]) at RT. Light field images were acquired using FSX100 (Olympus).

## Sulfo-NHS-LC-biotin perfusion in retina

Deeply anesthetized P15 mice were perfused with 0.75 μg/g (mouse weight) of Sulfo-NHS-LC-biotin (Thermo Fisher Scientific [21335]) dissolved in 10 ml PBS-CMF for 10 min, followed immediately by 10% formalin in sodium phosphate, pH7.4. Eyes were post-fixed in 4% PFA in PBS for 15 min on ice, washed with PBS, and soaked in 2xPBS for 15 min on ice. Dissected retina were flattened by dropping cold methanol and kept at −20°C. The flat retinas were washed with PBS, incubated with Perm/Block solution including 5% goat serum prior to immunostaining with CF488A-conjugated streptavidin (10 μg/ml) and Dylight594-conjugated IB4 (2.5 μg/ml) for 4 hr at RT or overnight at 4°C. After four washes with PBS, retinas were mounted and observed using confocal microscope A1R-TiE (Nikon).

## Isolation of murine brain and lung microvascular endothelial cells and qRT-PCR

Brain microvascular ECs were isolated as described previously (*Ruck et al., 2014*). In brief, 10 mice at P14 were sacrificed, and brains were isolated and transferred to 5 ml of PBS. Cerebellum and brainstem were removed with forceps. Meninges were detached by rolling the brains on sterile blotting paper. Meninges-free brains were transferred to a 50 ml Falcon tube filled with 13.5 ml of DMEM, minced first with a 25 ml pipette, then with a 10 ml pipette until the medium became milky. Tissue homogenates were digested by adding 0.6 ml of 10 mg/ml collagenase type 2 (Worthington [4174]) in DMEM and 0.2 ml of 1 mg/ml DNase I (Boehringer Mannheim [104159]) in PBS for 1 hr at 37°C on an orbital shaker at 180 rpm. After digestion, 10 ml of DMEM was added and the tissue

**Table 2.** Primers used for qRT-PCR.

| Target | Forward | Reverse |
|---|---|---|
| *Notch1* | AGTGTGACCCAGACCTTGTGA | AGTGGCTGGAAAGGGACTTG |
| *Notch2* | CCCAAGGACTGAGAGTCAGG | GGCAGCGGCAGGAATAGTGA |
| *Notch3* | ATTTGAGGGGTGCTGAAGTG | GAAGGCTGGGACAGAGAGAA |
| *Notch4* | TCCGGACTTTTAAGGCCAAA | TTCCCATTGCTGTGCATACTCT |
| *Dll4* | CCCACAATGGCTGTCGTCAT | AACCCTTTGGCCCACTGTTG |
| N-cadherin | AGGGTGGACGTCATTGTAGC | CTGTTGGGGTCTGTCAGGAT |
| CD31 | AGCCAACAGCCATTACGGTTA | AGCCTTCCGTTCTCTTGGTG |
| *GAPDH* | GGTGCTGAGTATGTCGTGGA | CCTTCCACCATGCCAAAGTT |
| *Jag1* | TCTCTGACCCCTGCCATAAC | TTGAATCCATTCACCAGATCC |
| *Hes1* | ACACCGGACAAACCAAAGAC | CGCCTCTTCTCCATGATAGG |
| *Hey1* | CATGAAGAGAGCTCACCCAGA | CGCCGAACTCAAGTTTCC |
| *Hey1* (mouse specific) | TGAATCCAGATGACCAGCTACTGT | TACTTTCAGACTCCGATCGCTTAC |
| *Vegfr1* | ACATTGGTGGTGGCTGACTCTC | CCTCTCCTTCGGCTGGCATC |
| *Vegfr2* | GCGGGCTCCTGACTACAC | CCAAATGCTCCACCAACTCTG |
| *Vegfr3* | CCGCAAGTGCATTCACAGAG | TCGGACATAGTCGGGGTCTT |

suspension was centrifuged at 1000 x *g* for 10 min at 4°C. The pellet was resuspended using a 25 ml pipette in 25 ml of 20% (w/v) BSA in DMEM approximately 25 times, and centrifuged at 1000 x g for 20 min at 4°C. The pellet was resuspended in 9 ml of DMEM and supplemented with 1 ml of 10 mg/ml collagenase type 2 and 0.1 ml of 1 mg/ml DNase. After digestion for 1 hr at 37°C, 15 ml of 20% FBS in DMEM containing penicillin/streptomycin was added to stop digestion. After centrifugation at 400x g for 5 min, the pellet was resuspended with 3 ml of 0.1% BSA in PBS, mixed with 22.5 µl of Dynabeads Biotin Binder (Veritas [11047]) precoated with 5 µg of biotin-labeled anti-CD31 antibody (clone 390; Biolegend [102404]). The mixture was incubated with rotation at RT for 15 min. Beads with bound ECs were collected using a magnet, washed with PBS, and subjected to total RNA isolation using PureLink RNA Mini Kit (Ambion [12183020]). Quantitative RT (qRT)-PCR was performed using M-MLV Reverse Transcriptase (Invitrogen [28025–013]) for reverse transcription, and SsoAdvanced Universal SYBR Green Supermix (Biorad [172–5271]) for real-time PCR analysis according to the manufacturer's instructions. Primers used for PCR are shown in *Table 2*.

Lung microvascular ECs were isolated as described previously (*Ruck et al., 2014*) unless otherwise noted. In brief, mice at 3 months were sacrificed, and lungs were minced and digested with 1 mg/ ml of collagenase/dispase (Roche [10 269 638 001]) in DMEM for 45 min at 37°C. A single cell suspension was passed through a 70 µm cell strainer, collected by centrifugation, and resuspended in 1 ml 0.1% BSA/PBS. ECs were isolated using Dynabeads Biotin Binder precoated with biotin-labeled anti-CD31 antibody (clone 390; Biolegend [102404]). Beads were resuspended in HuMedia-EG2 (Kurabou, [KE-2150S]) containing 2% FCS and 10 ng/ml hEGF, 5 ng/ml hFGF-b, 1.34 µg/ml hydrocortisone hemisuccinate, 10 µg/ml heparin, 50 µg/ml gentamycin, 50 ng/ml Amphotericin B, and then, plated on CellBIND Surface Culture Dish (Corning, [3295]). After reaching 70–80% confluent, cells were detached using 0.05% Trypsin/EDTA, collected by centrifugation, and resuspended in 1 ml 0.1% BSA/PBS. ECs were subsequently purified using Dynabeads Biotin Binder precoated with biotin-labeled anti-CD102 antibody (clone 3C4; Biolegend [105604]). Beads were resuspended with complete HuMedia-EG2 media, and cultured onto CellBIND Surface Culture Dish.

## Acknowledgements

We thank T Honjo for providing *Rbpj* mice, M Yanagisawa and RIKEN BioResource Center for providing *Tek-Cre* mice, K Uchimura for collaborative work, N Hattori for technical assistance, K Furukawa for supporting and supervising the project, M Nakakura and T Kawai for experiments in the initial phase of the project.

# Additional information

## Funding

| Funder | Grant reference number | Author |
| --- | --- | --- |
| Japan Society for the Promotion of Science | JP15K15064 | Mitsutaka Ogawa<br>Tetsuya Okajima |
| Takeda Science Foundation | | Tetsuya Okajima |
| Japan Foundation for Applied Enzymology | | Tetsuya Okajima |
| National Institutes of Health | RO1 GM106417 | Pamela Stanley |
| Japan Society for the Promotion of Science | JP26110709 | Tetsuya Okajima |
| Japan Society for the Promotion of Science | JP26291020 | Kyosuke Takeshita<br>Tetsuya Okajima |
| Japan Society for the Promotion of Science | JP15K18502 | Mitsutaka Ogawa |
| Japan Society for the Promotion of Science | JP16J00004 | Mitsutaka Ogawa |
| Japan Society for the Promotion of Science | JP15K07935 | Hirokazu Yagi |
| Japan Society for the Promotion of Science | JP26110716 | Hirokazu Yagi |
| Japan Society for the Promotion of Science | JP25102008 | Koichi Kato |

The funders had no role in study design, data collection and interpretation, or the decision to submit the work for publication.

## Author contributions

SSa, Conceptualization, Resources, Data curation, Investigation, Methodology, Writing—original draft, Writing—review and editing, Performed and interpreted in vitro experiments, Collected data for Eogt-deficient, Notch1+/- and Rbpj+/- compound mice; SV, Conceptualization, Resources, Data curation, Investigation, Methodology, Writing—original draft, Writing—review and editing, Performed and interpreted in vitro experiments, Collected data for Eogt-deficient, Notch1 12f and Notch1 lbd compound mice; MO, Conceptualization, Resources, Data curation, Funding acquisition, Investigation, Methodology, Writing—original draft, Writing—review and editing, Collected data for Eogt-deficient, Notch1+/- and Rbpj+/- compound mice, Obtained funding; YS, Resources, Investigation, Writing—review and editing, Generated and analyzed Eogt-deficient mice, Assisted with preparing the manuscript; HY, KK, Resources, Funding acquisition, Methodology, Writing—review and editing, Contributed unpublished materials and information, Assisted with preparing the manuscript, Obtained funding; KT, Resources, Funding acquisition, Methodology, Writing—review and editing, Contributed materials and interpretation of data, Assisted with preparing the manuscript, Obtained funding; TM, Resources, Methodology, Writing—review and editing, Contributed materials and interpretation of data, Assisted with preparing the manuscript; SSu, Resources, Methodology, Writing—review and editing, Performed and interpreted in vitro experiments, Assisted with preparing the manuscript; PS, TO, Conceptualization, Data curation, Supervision, Funding acquisition, Writing—original draft, Project administration, Writing—review and editing, Supervised the project, Obtained funding

## Author ORCIDs

Pamela Stanley, http://orcid.org/0000-0001-5704-3747
Tetsuya Okajima, http://orcid.org/0000-0002-3677-648X

## Ethics

Animal experimentation: All experimental procedures were performed in accordance with the Guidelines for Animal Experimentation in Nagoya University Graduate School of Medicine and Japanese Government Animal Protection and Management Law (Permit Number: 26397). Animal experiments at the Albert Einstein College of Medicine were performed with the approval of the Institutional Animal Care and Use Committee (Permit Number: 20140803).

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
