## [Decision Letter]

[Editors’ note: a previous version of this study was rejected after peer review, but the authors submitted for reconsideration. The first decision letter after peer review is shown below.]

Thank you for submitting your work entitled "O-GlcNAc on NOTCH1 EGF Repeats Regulates Ligand-Induced Notch Signaling and Vascular Development in Mammals" for consideration by *eLife*. Your article has been reviewed by two peer reviewers, and the evaluation has been overseen by a Reviewing Editor and Janet Rossant as the Senior Editor. The following individuals involved in review of your submission have agreed to reveal their identity: Holger Gerhardt (Reviewer #2).

Our decision has been reached after consultation between the reviewers. Based on these discussions and the individual reviews below, we regret to inform you that your work will not be considered further for publication in *eLife*.

The reviewers were supportive of the overall concept that O-GlcNacylation could be another form of glycosylation with impact on Notch signaling. However, in their reviews and in the online discussion between the editor and the reviewers, there was concern that the in vivo phenotype reported was very mild and not fully substantiated. Also, the in vitro demonstration of EOGT function in modulating Notch signaling used cell lines that were not related to the in vivo endothelial function. It was felt that this study needs a complete revision with careful analysis of EOGT biochemistry and effects on Notch signaling in endothelial cells in vitro, and a much better characterization of the in vivo phenotypes. As it stands, this work does not provide the same level of rigorous validation provided by the published papers on the effects of other glycosylation enzymes in Notch biology. Because the paper will need substantial additional work to achieve the required quality for publication in *eLife*, we are unable to accept the manuscript at this time. We hope that the referees' comments will help you improve the paper for future publication.

*Reviewer #1:*

Sawaguchi and colleagues present a rigorous study that describes the regulatory role of EGF O-GlcNAc transferase (EOGT) in the mammalian Notch signaling pathway. Previous studies from Okajima and Stanley laboratories reported that loss of eogt in flies does not impair Notch signaling but shows a dosage-sensitive genetic interaction with components of the Notch pathway, suggesting that eogt might be a modulator of the Notch pathway. More recently, loss-of-function mutations of EOGT were identified in an autosomal recessive form of Adams-Oliver Syndrome (AOS), a disease which is also caused by dominant mutations in several Notch pathway components. Despite these observations, to my knowledge, no functional links had been established between Eogt and mammalian Notch signaling. To address this issue, the authors generated germline and conditional Eogt alleles and showed that although Eogt^-/-^ animals are viable, they exhibit phenotypes compatible with impaired Notch signaling during retinal angiogenesis and genetically interact with Notch pathway components during this process. Conditional loss of Eogt in endothelial cells indicated that the observed phenotypes are primarily, although not exclusively, caused by loss of Eogt from endothelial cells. The authors complement their in vivo observations with a number of cell-based assays and provide strong evidence that the vascular phenotype observed in the EOGT mutants is mediated through its effect on Notch1-DLL4 binding and signaling. Of note, EOGT does not seem to affect JAG1-Notch1 binding and signaling. Finally, by introducing mutations in four conserved O-GlcNAc sites outside of the core ligand-binding domain of Notch1, the authors show that O-GlcNAc moieties on specific EGF repeats of mammalian Notch1 likely enhance Notch1-DLL4 interaction. It is somewhat surprising that EOGT-null mice fail to recapitulate any of the gross clinical pathologies observed in AOS patients, and it might be a genetic background issue. Nevertheless, the study proves that EOGT is a regulator of the Notch1 signaling during mouse angiogenesis and indicates that addition of O-GlcNAc to the Notch1 receptor promotes its binding to and signaling in response to DLL4 but not JAG1.

In my opinion, the manuscript addresses an important biological question, with potential implications to a human disease. The authors provide mechanistic insight into the regulation of mammalian Notch signaling by EOGT. The major conclusions of the study are well supported by the data. It might have been better to introduce T-to-V mutations in EGF repeats 10 and 17 instead of T-to-A, as T is more similar to V than to A, and therefore the chance for seeing a result due to altered amino acid structure rather than lack of sugar might be somewhat higher in T-to-A mutation. However, the severe decrease in O-GlcNAc Western blot band on mutant Notch1 and all the other data in the manuscript do not leave much doubt that the observed effects are due to the loss of O-GlcNAc saccharides.

Reviewer #1 (Minor Comments):

Specific issues:

1) Introduction, second paragraph: The reference "Shaheen, 2011" seems out of place here since the cited article does not mention EOGT. It seems that the authors/word-processor may have inadvertently swapped "Shaheen, 2013" from earlier in the paragraph, with this one.

2) Subsection “Loss of Eogt phenocopies Notch1 haploinsufficiency and compound mutants display an enhanced vascular phenotype”, last paragraph: Please rewrite these sentences to address the following issues: (1) The authors write "data not shown", but they have shown quantifications for the hypomorphic alleles in 3B and 3C. (2) I do not agree with the authors' interpretation of the genetic interaction studies. First, in the case of Eogt interaction with Rbpj+/- and N1+/-, both components of the genetic interaction have a phenotype by themselves, one functional copy of Rbpj and N1 remain, and the interaction is not synergistic. Therefore, it is hard to draw conclusions. If Eogt and N1/Rbpj acted in parallel and independently of each other, the authors might still have seen similar results. Second, the results with the hypomorphic alleles are stronger and strongly suggest that EOGT regulates Notch pathway in this context (because they don't have a phenotype by themselves). However, it does not inform us about the actual target of EOGT in the pathway. Even if O-GlcNAc residues on DLL4 were important, a similar genetic interaction might have been observed. Therefore, it cannot be concluded that "EOGT modification of NOTCH1 is important in regulating the Notch signaling pathway."

3) Subsection “Eogt regulates Notch pathway target genes in ECs and ligand-induced Notch signaling”, first paragraph: "These results suggested that EOGT affects Notch signaling mediated by DLL4" – although this is a reasonable conclusion at the end of the paper, the results presented up to this point only suggest that Notch signaling is suppressed but do not seem to provide an indication that the effect is limited to one particular ligand.

4) Subsection “Eogt regulates Notch pathway target genes in ECs and ligand-induced Notch signaling”, third paragraph: an siRNA.

5) Subsection “EOGT regulates DLL1 and DLL4 binding to NOTCH1”: In Figure 6, it looks like DLL4 binding is slightly decreased. But in Figure 6, the decrease does not seem to be statistically significant. Please clarify.

6) Subsection “O-GlcNAcylation of NOTCH1 promotes DLL4-Notch1 interaction”, first paragraph: "Thus it appears that multiple sites are O-GlcNAcylated in NOTCH1." Although this conclusion is correct, I think an equally if not more important conclusion is that the consensus is probably too wide. Notch1 has more than 20 predicted sites or 11 sites based on the more stringent consensus sequence. However, mutating 4 sites almost abolished the O-GlcNAc band in Western blot (Figure 7—figure supplement 1). This strongly suggests that only some sites are modified.

7) Discussion, first paragraph: The logic of this section is not clear to me. Yes, removing one enzyme should not affect other sugars, but I don't understand how this leads to the last sentence. I suggest that instead of these sentences, the authors add a few sentences about the previous two studies on the fly eogt from their labs, mention that eogt null flies did not have Notch phenotypes, and eogt RNAi in the wing showed some genetic interaction with Notch pathway components and then end by what they have. In my opinion, this should not affect the novelty of their work at all, but provides a better framework for the paper. It will also serve as an example to the readers that when a posttranslational modification regulates mammalian Notch signaling in a tissue-specific manner, it might not show overt Notch loss of function phenotypes in *Drosophila*.

8) Discussion, second paragraph: "DLL4 constitutes the critical Notch ligand in regulating angiogenesis and vascular development". This sentence gives the impression that DLL4 is the only important Notch ligand in vascular development. However, JAG1 is also important in vascular development. Please modify to make the sentence more accurate.

9) Discussion, third paragraph: The authors have previously reported that at least in mammalian cells, O-GlcNAc on EGF repeats can be extended by the addition of a galactose. Given that in this paragraph the authors highlight some similarities between the functions of Eogt and Fringe proteins, and because Fringe adds the second sugar not the first one, it is tempting to speculate that the Eogt phenotype might result from the loss of galactose. Of note, to my knowledge, this elongation has not been observed in flies (for example, see Harvey et al., JBC, 2016). Therefore, this hypothesis can also explain why eogt mutant flies do not show any obvious Notch loss of function phenotypes. It will be up to the authors whether they want to incorporate this hypothesis in their Discussion or not.

10) Discussion, third paragraph: "JAG1-Notch1 interactions were not evident in our assays employing Eogt-deficient.…" Instead of evident, I think affected or something like that should be used.

11) Discussion, last paragraph: "we generated mice homologous for Eogt gene". Homozygous?

12) Subsection “Materials and mice”, first paragraph: It is hard for the reader to understand from the rest of the article exactly which Cre mouse line was used to generate the global KO (I'm assuming it was Ayu1-Cre). I suggest rewriting this sentence more clearly. Also, in the first paragraph of the subsection “Generation of Eogt-targeted mice” the authors use "Alternatively" to describe the generation of global knockout mice. Do they mean "In addition"?

13) Figure 1: Please show a scale bar for the schematic so that the predicted length of the Southern blot bands can be estimated.

14) Figure 5: Please provide labels for wt versus mutant in the figure.

*Reviewer #2:*

The biology of Notch signalling in development and disease, and in vascular morphogenesis in particular, is highly relevant and of broad general interests. Understanding the multiple layers of transcriptional, translational and post-translational regulation of Notch signalling is a major challenge, and the present work adds by addressing the role of O-GlcNacylation via the enzyme EOGT.

The authors study effects of general or endothelial specific loss of function using mouse mutants and the model of post-natal retinal vascularisation. The same model has been extensively studied and the effects of loss of function of the ligands Dll1, Dll4, Jagged1, the receptors Notch1, and various modifiers as well as signalling components including RBPJ are very well understood. The authors here demonstrate a very mild hypervascularization, claiming similarities to other Notch mutants. In addition, the general EOGT knockout shows reduced aSMA staining indicating deficient smooth muscle coverage. Furthermore, the authors claim reduced vessel integrity. The authors then move on to investigate the mechanism and identify reduced Dll1 or Dll4 mediated Notch activation, correlating with reduced Notch O-GlcNAcylation. Various cell lines including Hela, HEK, CHO are used for an array of assays to confirm reduced signalling, ligand binding and investigate whether Dll4 and Jagged1 show similarly reduced binding when EOGT is deficient. Finally, the authors show by mutation of the putative glycosylation sites the these indeed affect Dll4, but not Jagged1.

Overall, this is a potentially interesting piece of work, mostly clearly written and concise. However, I have several major concerns that reduce my confidence in the conclusions and in the general relevance of the findings. The retinal analysis shows a very mild if any phenotype, and this is not at all comparable to other pathway mutants.

The quality of the images, magnifications, microscopy technique etc. undermine the ability to appreciate what the authors claim. For example, the filopodia quantification images do not allow visualisation of filopodia therefore making it difficult to appreciate the quantifications.

The same is true for the leakage data. This is not a standard BRB assay and what the authors show as leakage cannot be appreciated in those images. Why would this be focal? Not diffuse?

Why do the authors then choose Hela, HEK and CHO cells for functional studies, with the odd choice of GAPDH shRNA as control?

The WB panels in Figure 5 are oddly assembled with mismatch between the grey line and the actual gel cuts. I am not suggesting this is attempted image manipulation, but it is either not careful enough or indeed indicative of multiple gels that are assembled together. This needs clarification.

The evidence of Notch1 O-GlcNacylation in Figure 5 also seems over interpreted as co-migration is not evident.

The authors also use 3 different GSI, but do not mention why and where in the text.

The binding assay with beads needs Notch null controls as the lack of O-GlcNacylation will not be Notch specific and other effects might interfere with the adherent of the beads.

Overall, although I am generally positive about the topic and the ideas in the work, I feel this study needs a complete revision with careful analysis of EOGT biochemistry and effects in endothelial cells in vitro, and a much better illustration of the in vivo effects.

Also, whether this is Notch1 selective or generally affects Notch receptor affinities, importantly also in vascular smooth muscle, needs to be addressed. Furthermore, as also ligands have these EGF repeats, it would seem important to include analysis of potential effects here. As it stands, this work is not equally convincing as the published effects of other glycosylation enzymes in Notch biology and requires substantial additional work to achieve the required quality. Given the policy of *eLife* for revisions, I feel that the extent of work required to clarify and improve is beyond what would constitute a normal revision.

[Editors’ note: what now follows is the decision letter after the authors submitted for further consideration.]

Thank you for resubmitting your work entitled "O-GlcNAc on NOTCH1 EGF Repeats Regulates Ligand-Induced Notch Signaling and Vascular Development in Mammals" for further consideration at *eLife*. Your article has been favorably evaluated by Fiona Watt (Senior Editor) and three reviewers, one of whom is a member of our Board of Reviewing Editors.

The manuscript has been improved but there are some remaining issues that need to be addressed before acceptance, as outlined below:

The reviewers appreciate the value of restructuring the manuscript and data organisation, and believe the core conclusions are sufficiently supported by your new experiments. They also state your work exposes an interesting contrast between the functions of Fringe proteins and EOGT. Previous studies showed that addition of GlcNAc to O-linked fucose on Notch1 by Fringe proteins promotes DLL-induced signaling and simultaneously reduces JAG1-mediated signaling. However, your work identifies that addition of O-linked GlcNAc to Notch1 by EOGT seems to promote DLL-Notch1 binding and signaling without reducing JAG1-Notch1 binding. Therefore, although the EOGT loss of function phenotypes in vivo are rather weak, the reviewers agree that linking the function of EOGT to mammalian Notch signaling for the first time and this nuance in ligand-specific activity of EOGT will make your paper of significant interest to *eLife* readership.

Having said that, we ask you to make the following minor revisions, which should not necessarily require further experiments:

1) In Figure 3, the western blots showing EOGT expression are poor. It is difficult to discern endogenous EOGT band, because it is obscured by a non-specific band. It is important to corroborate the (extent of) ablation of EOGT expression in the shRNA-expressing cell lines, but it is difficult to evaluate this in the present data.

2) In Figure 3, the cells are indicated to stably express siRNAs. Should this be shRNAs? Please clarify/correct.

3) We recommend to revise Figure 1 based on Rana et al. (JBC, 2011), which reported the identification of O-glucose glycans on 17 EGF repeats of the mouse Notch1.

4) Subsection “Eogt regulates Notch pathway target genes in ECs and ligand-induced Notch signaling”: "In contrast, Eogt-/- ECs showed a significant decrease in the induction of Notch target genes stimulated by DLL4, but not JAG1 (Figure 8)." It might be more accurate to say that "DLL4 fails to induce Notch target genes Hey1 and Hes1 in Eogt-/- endothelial cells, but JAG1 results in a robust induction of these genes".

5) You cite a proteomic study (Alfaro et al., PNAS, 2012) that identified a cohort of membrane proteins modified by GlcNac. One of these proteins identified was *Notch2*. It would be useful to add a discussion on this to the manuscript: can EOGT-mediated modification of *Notch2* explain the phenotypes observed in the current manuscript?

---

## [Author Response]

[Editors’ note: the author responses to the first round of peer review follow.]

We note that Reviewer #1 concludes “Nevertheless, the study proves that EOGT is a regulator of the Notch1 signaling during mouse angiogenesis….” and that “the major conclusions are well supported by the data”. We thank this reviewer for pointing out certain discrepancies that need clarification, which we have addressed as follows:

Reviewer #1 (Minor Comments):

*Specific issues:*

*1) Introduction, second paragraph: The reference "Shaheen, 2011" seems out of place here since the cited article does not mention EOGT. It seems that the authors/word-processor may have inadvertently swapped "Shaheen, 2013" from earlier in the paragraph, with this one.*

Corrected (Introduction, second paragraph).

*2) Subsection “Loss of Eogt phenocopies Notch1 haploinsufficiency and compound mutants display an enhanced vascular phenotype”, last paragraph: Please rewrite these sentences to address the following issues: (1) The authors write "data not shown", but they have shown quantifications for the hypomorphic alleles in 3B and 3C. (2) I do not agree with the authors' interpretation of the genetic interaction studies. First, in the case of Eogt interaction with Rbpj+/- and N1+/-, both components of the genetic interaction have a phenotype by themselves, one functional copy of Rbpj and N1 remain, and the interaction is not synergistic. Therefore, it is hard to draw conclusions. If Eogt and N1/Rbpj acted in parallel and independently of each other, the authors might still have seen similar results. Second, the results with the hypomorphic alleles are stronger and strongly suggest that EOGT regulates Notch pathway in this context (because they don't have a phenotype by themselves). However, it does not inform us about the actual target of EOGT in the pathway. Even if O-GlcNAc residues on DLL4 were important, a similar genetic interaction might have been observed. Therefore, it cannot be concluded that "EOGT modification of NOTCH1 is important in regulating the Notch signaling pathway."*

The text is revised to modify the interpretation of genetic interaction data as recommended, by explaining that enhancement of heterozygous *Notch1* and *Rbpj* phenotypes in *Eogt*[-/-] retinas could reflect additive effects of independent pathways, whereas enhancement observed in *Notch1*[12f] or *Notch1*[lbd] backgrounds, that lack a heterozygous phenotype, is genetic evidence that it is the loss of O-GlcNAc on NOTCH1 which is responsible for the enhanced phenotype (subsection “Loss of Eogt phenocopies Notch1 haploinsufficiency and compound mutants display an enhanced vascular phenotype”).

*3) Subsection “Eogt regulates Notch pathway target genes in ECs and ligand-induced Notch signaling”, first paragraph: "These results suggested that EOGT affects Notch signaling mediated by DLL4" – although this is a reasonable conclusion at the end of the paper, the results presented up to this point only suggest that Notch signaling is suppressed but do not seem to provide an indication that the effect is limited to one particular ligand.*

The reorganized Results section now presents the DLL4-NOTCH1 binding data before the mouse data.

*4) Subsection “Eogt regulates Notch pathway target genes in ECs and ligand-induced Notch signaling”, third paragraph: an siRNA.*

Corrected (subsection “Knockdown of EOGT reduces Notch signaling”, last paragraph).

*5) Subsection “EOGT regulates DLL1 and DLL4 binding to NOTCH1”: In Figure 6, it looks like DLL4 binding is slightly decreased. But in Figure 6, the decrease does not seem to be statistically significant. Please clarify.*

We reviewed all our ligand binding data and now include data obtained with different concentrations of ligands. Panel C of revised Figure 1 compares control and Eogt KD cells and Panel D shows data for the three experiments in which rescue of the knockdown phenotype was performed.

*6) Subsection “O-GlcNAcylation of NOTCH1 promotes DLL4-Notch1 interaction”, first paragraph: "Thus it appears that multiple sites are O-GlcNAcylated in NOTCH1." Although this conclusion is correct, I think an equally if not more important conclusion is that the consensus is probably too wide. Notch1 has more than 20 predicted sites or 11 sites based on the more stringent consensus sequence. However, mutating 4 sites almost abolished the O-GlcNAc band in Western blot (Figure 7—figure supplement 1). This strongly suggests that only some sites are modified.*

The new western analysis (Figure 2) confirmed the point raised by the reviewer that O-GlcNAcylation occurs on a limited number of EGF repeats. We have revised the manuscript accordingly (end of subsection “O-GlcNAc on NOTCH1 promotes DLL4-NOTCH1 interactions”). We also added a paragraph in the Discussion (fifth paragraph) and updated Figure 2—figure supplement 1, accordingly.

*7) Discussion, first paragraph: The logic of this section is not clear to me. Yes, removing one enzyme should not affect other sugars, but I don't understand how this leads to the last sentence. I suggest that instead of these sentences, the authors add a few sentences about the previous two studies on the fly eogt from their labs, mention that eogt null flies did not have Notch phenotypes, and eogt RNAi in the wing showed some genetic interaction with Notch pathway components and then end by what they have. In my opinion, this should not affect the novelty of their work at all, but provides a better framework for the paper. It will also serve as an example to the readers that when a posttranslational modification regulates mammalian Notch signaling in a tissue-specific manner, it might not show overt Notch loss of function phenotypes in Drosophila.*

We have clarified the logic in the first paragraph of the Introduction. We also added a few sentences as suggested by the reviewer in the Discussion (fifth paragraph).

*8) Discussion, second paragraph: "DLL4 constitutes the critical Notch ligand in regulating angiogenesis and vascular development". This sentence gives the impression that DLL4 is the only important Notch ligand in vascular development. However, JAG1 is also important in vascular development. Please modify to make the sentence more accurate.*

We have modified the revised text to take this point into account (Discussion, first paragraph).

*9) Discussion, third paragraph: The authors have previously reported that at least in mammalian cells, O-GlcNAc on EGF repeats can be extended by the addition of a galactose. Given that in this paragraph the authors highlight some similarities between the functions of Eogt and Fringe proteins, and because Fringe adds the second sugar not the first one, it is tempting to speculate that the Eogt phenotype might result from the loss of galactose. Of note, to my knowledge, this elongation has not been observed in flies (for example, see Harvey et al., JBC, 2016). Therefore, this hypothesis can also explain why eogt mutant flies do not show any obvious Notch loss of function phenotypes. It will be up to the authors whether they want to incorporate this hypothesis in their Discussion or not.*

We have included this point raised by the reviewer and also noted that B4GALT1 knockout mice exhibit only a mild reduction in the expression of some Notch target genes during somitogenesis in which O-fucose glycans extended by Lfng and GalT are essential for skeletal and tail development (Discussion, fourth paragraph). There is no obvious skeletal or tail phenotype in Eogt[-/-] mice.

*10) Discussion, third paragraph: "JAG1-Notch1 interactions were not evident in our assays employing Eogt-deficient.…" Instead of evident, I think affected or something like that should be used.*

Corrected (Discussion, second paragraph).

*11) Discussion, last paragraph: "we generated mice homologous for Eogt gene". Homozygous?*

Corrected (Discussion, last paragraph).

*12) Subsection “Materials and mice”, first paragraph: It is hard for the reader to understand from the rest of the article exactly which Cre mouse line was used to generate the global KO (I'm assuming it was Ayu1-Cre). I suggest rewriting this sentence more clearly. Also, in the first paragraph of the subsection “Generation of Eogt-targeted mice” the authors use "Alternatively" to describe the generation of global knockout mice. Do they mean "In addition"?*

Corrected in Methods (subsection “Mice”). “Alternatively” is correct and has not been changed.

*13) Figure 1: Please show a scale bar for the schematic so that the predicted length of the Southern blot bands can be estimated.*

Included (Figure 4).

*14) Figure 5: Please provide labels for wt versus mutant in the figure.*

Labels added (Figure 8).

*Reviewer #2:*

*[…] Overall, this is a potentially interesting piece of work, mostly clearly written and concise. However, I have several major concerns that reduce my confidence in the conclusions and in the general relevance of the findings. The retinal analysis shows a very mild if any phenotype, and this is not at all comparable to other pathway mutants.*

The Notch pathway comprises Notch receptors, Notch ligands, enzymes such as presenilin, Pofut1 and Fringe, and transcription modulators including RBP-Jk and Mastermind. Conditionally removing NOTCH1, DLL4 or RBP-Jk has a dramatic effect on retinal angiogenesis but removing Lunatic Fringe gives a mild phenotype (Benedito et al. 2009). Since the latter report, the consequences of removing any other glycosyltransferase in the Notch pathway have not been published. As explained in the revised text, our experiments removed only one of three classes of O-glycans, so we were expecting a very mild phenotype. Both O-fucose and O-glucose glycans are important in promoting Notch signaling in many contexts, including angiogenesis in the case of the O-fucose glycans that are extended by LFNG. Manic and Radical Fringe are also expressed in retina but retinal angiogenesis experiments in *Mfng*[-/-] or *Rfng*[-/-] mice have not been reported. We have examined *Eogt*[-/-] retinas from numerous mice at P5 and P15 and found significant differences from wild type retinas in several parameters. The combined data convincingly demonstrate changes consistent with reduced Notch signaling. We have included a diagram of the predicted O-glycans based on known consensus sequences in the extracellular domain of mouse Notch1 to highlight the point that knocking out EOGT removes only one of three types of O-glycan (Figure 1).

*The quality of the images, magnifications, microscopy technique etc. undermine the ability to appreciate what the authors claim. For example, the filopodia quantification images do not allow visualisation of filopodia therefore making it difficult to appreciate the quantifications.*

We have worked with our analytical imaging facilities to improve all images in the manuscript. In particular, the images of filopodia which showed how they were counted now clearly show the increased numbers generated in *Eogt*[-/-] retinas.

*The same is true for the leakage data. This is not a standard BRB assay and what the authors show as leakage cannot be appreciated in those images. Why would this be focal? Not diffuse?*

As pointed out by the reviewer, the leakage of sulfo-NHS-LC-biotin is diffusive in the case of a severe phenotype. However, in the case of a milder phenotype, it looks focal rather than diffuse. Please refer to the following literature: Figure 1 (SIRS vs. SIRS+2-BP), Richard S. Beard Jr. et al. Nature Communications: 12823 (2016) doi:10.1038/ncomms12823. Non-diffusive sulfo-NHS-biotin staining in retinal vasculature was also reported (Xuwen Liu Am. J. Pathol. 2016). Sulfo-NHS-LC-biotin reacts with proteins that have primary amines. Therefore, a small amount of sulfo-NHS-LC-biotin, once it is extravasated, will be easily trapped in the extracellular matrix or on cell surface proteins. That may be the reason the staining patterns differ depending on the amount of the tracers leaked from vessels. To clarify the extravascular sulfo-NHS-LC-biotin staining, we added 3D images in higher resolution (Figure 7) and moved the original images to Figure 7—figure supplement 1. As an independent measure for increased vascular permeability, we performed staining for fibrinogen, as analyzed in Notch3-/- retinas (Henshall TL et al. Arterioscler Thromb Vasc Biol. (2015) 35:409-420). In Eogt HO retinas, the staining looks more diffuse (Figure 7). Similar extravascular staining was also observed in Eogt EC KO, Notch1 HT, and Rbpj HT retinas (Figure 7).

*Why do the authors then choose Hela, HEK and CHO cells for functional studies, with the odd choice of GAPDH shRNA as control?*

The revised manuscript explains our rationale. We began our studies with

HeLa, CHO and HEK-293T cells to determine if we could see effects of knocking down or knocking out *Eogt* in cell-based assays. We note that Benedito et al. (2009) also used HeLa cells to investigate Fringe effects in co-culture Notch signaling assays. Also, it was important to show that effects we saw were not confined to a single cell line. The combined experiments provide comprehensive data that reducing EOGT causes a reduction in Notch1 activation (HeLa), reduced Notch1 signaling (CHO) and reduced Notch1 binding to Delta ligands, but no change to Jagged1 binding (CHO and HEK-293T). We also isolated endothelial cells (EC) from *Eogt*[-/-] mouse brain and lung and showed that Notch pathway target gene transcripts are reduced in *Eogt*[-/-] versus *Eogt*[+/+] ECs (see new Figure 8).

The technical concerns of the reviewer have been addressed as follows:

*The WB panels in Figure 5 are oddly assembled with mismatch between the grey line and the actual gel cuts. I am not suggesting this is attempted image manipulation, but it is either not careful enough or indeed indicative of multiple gels that are assembled together. This needs clarification.*

The western blots in revised Figure 3 were performed on sections of PVDF membrane cut from a single membrane and probed with the relevant antibody. The panel probed for Notch1 was stripped and probed for O-GlcNAc. The single membrane was then reassembled for the figure. We now present the panels separately and explain more clearly how the western blots were probed.

*The evidence of Notch1 O-GlcNacylation in Figure 5 also seems over interpreted as co-migration is not evident.*

Co-migration was evident from the markers on the original gels which are now included in the figure.

*The authors also use 3 different GSI, but do not mention why and where in the text.*

We used different GSIs for no particular reason other than they were on hand in our two laboratories. It is now made clear in the figure legends which GSI was used in each experiment. L-685,458 was used in some experiments but not those shown.

*The binding assay with beads needs Notch null controls as the lack of O-GlcNacylation will not be Notch specific and other effects might interfere with the adherent of the beads.*

We now included NOTCH1-null control data in Figure 1. The fact that Notch1 transfection increases JAG1- and DLL4-bead binding provides good evidence that our assay indeed measures NOTCH1 recognized by DLL4 and JAG1. In contrast to *Notch1*-null cells, DLL4 binding but not JAG1 binding was markedly diminished in *Notch1*- transfected *Eogt*-null cells. Similarly, removal of O-GlcNAc sites in Notch1 decreased DLL4 binding. These data clearly show that the specific effect of O-GlcNAc on NOTCH1 which affects DLL4-NOTCH1 binding.

*Overall, although I am generally positive about the topic and the ideas in the work, I feel this study needs a complete revision with careful analysis of EOGT biochemistry and effects in endothelial cells in vitro, and a much better illustration of the in vivo effects.*

Now the manuscript is completely revised by performing additional analysis of the effect of EOGT on endothelial cells (Figure 8), including the improved biochemical data for Notch1 O-GlcNAcylation by EOGT (Figure 3), improved images (e.g. Figure 5, Figure 7), additional data on retinas (e.g. Figure 7), and better illustration of in vivoeffects in the Results and Discussion.

*Also, whether this is Notch1 selective or generally affects Notch receptor affinities, importantly also in vascular smooth muscle, needs to be addressed. Furthermore, as also ligands have these EGF repeats, it would seem important to include analysis of potential effects here. As it stands, this work is not equally convincing as the published effects of other glycosylation enzymes in Notch biology and requires substantial additional work to achieve the required quality. Given the policy of eLife for revisions, I feel that the extent of work required to clarify and improve is beyond what would constitute a normal revision.*

Eogt[-/-] retinas were analyzed for vascular smooth muscle cells and pericytes by staining with aSMA and NG2 antibodies in P15. The results revealed that the coverage with vascular smooth muscle and pericyte investment are unaffected (Figure 7), which suggests that cellular basis for impaired vascular integrity is distinct from Notch3[-/-] retina (subsection “EOGT is required for retinal vascular integrity”, last paragraph) and thus Notch3 appears to function without Eogt in this context. As also suggested by the reviewer, we analyzed the requirement for EOGT in ligand function. EOGT-null cells were transfected to express Notch ligands and their ability to send signals was compared with wild-type cells. The results showed that EOGT is dispensable for the ability of DLL4 and JAG1 to send signals (Figure 8).

[Editors’ note: the author responses to the re-review follow.]

*[…] Having said that, we ask you to make the following minor revisions, which should not necessarily require further experiments:*

*1) In Figure 3, the western blots showing EOGT expression are poor. It is difficult to discern endogenous EOGT band, because it is obscured by a non-specific band. It is important to corroborate the (extent of) ablation of EOGT expression in the shRNA-expressing cell lines, but it is difficult to evaluate this in the present data.*

As is often the case with glycosyltransferases, endogenous EOGT levels are too low to be readily detected by Western analysis in CHO and HeLa cell lysates. Similarly, it has been shown that levels of glycosyltransferase gene transcripts may not reflect protein levels or enzyme activity. Thus, we decided that the best test of EOGT knockdown was reduction in

O-GlcNAc (the product of EOGT activity), as detected by the anti-O-GlcNAc Ab CTD110.6. Western blots of each construct are shown in Figure 9. All constructs showed reduced O-GlcNAc signal, consistent with the data from an independent experiment shown in Figure 3 for TA197 and TA198 shRNAs. The latter constructs were the most effective at reducing ligand-induced NOTCH1 activation as shown in Figure 3, and so subsequent experiments were pursued with these two constructs. This explanation has been included in the revised text (subsection “Knockdown of EOGT reduces Notch signaling”, first paragraph).

Author response image 1.**DOI:**
http://dx.doi.org/10.7554/eLife.24419.026

Characterization of *Eogt*-siRNA CHO cells were published in Tashima and Stanley, JBC, (2014) as mentioned in the last paragraph of the aforementioned subsection.

2) In Figure 3, the cells are indicated to stably express siRNAs. Should this be shRNAs? Please clarify/correct.

Figure 3 reports experiments with Lec1 CHO cells stably expressing siRNA against CHO *Eogt* as stated in the figure legend. The characterization of these siRNA knockdown cells was reported in Tashima and Stanley, JBC, 2014 as noted in the Methods.

*3) We recommend to revise Figure 1 based on Rana et al. (JBC, 2011), which reported the identification of O-glucose glycans on 17 EGF repeats of the mouse Notch1.*

Changes were made as suggested.

*4) Subsection “Eogt regulates Notch pathway target genes in ECs and ligand-induced Notch signaling”: "In contrast, Eogt-/- ECs showed a significant decrease in the induction of Notch target genes stimulated by DLL4, but not JAG1 (Figure 8)." It might be more accurate to say that "DLL4 fails to induce Notch target genes Hey1 and Hes1 in Eogt-/- endothelial cells, but JAG1 results in a robust induction of these genes".*

Changes were made as suggested (subsection “Eogt regulates Notch pathway target genes in ECs and ligand-induced Notch signaling”, last paragraph).

*5) You cite a proteomic study (Alfaro et al., PNAS, 2012) that identified a cohort of membrane proteins modified by GlcNac. One of these proteins identified was Notch2. It would be useful to add a discussion on this to the manuscript: can EOGT-mediated modification of Notch2 explain the phenotypes observed in the current manuscript?*

NOTCH1 and NOTCH4 have been implicated in angiogenesis and vasculogenesis, but not *NOTCH2* to our knowledge. We have revised the sentences in Discussion as follows: “In fact, all Notch receptors are potentially modified with O-GlcNAc and *NOTCH2* was reported to be O-GlcNAcylated in mouse cerebrocortical tissue (Alfaro et al., 2012). Although roles for *Notch2* in vascular morphogenesis are not reported, *Notch1* and *Notch4* have partially overlapping roles (Krebs et al., 2000).” The potential contribution of O-GlcNAc in NOTCH4 regulation is also discussed (Discussion, third paragraph).